# High-resolution, genome-wide mapping of positive supercoiling in chromosomes

**Monica S Guo[1†]\*, Ryo Kawamura[2], Megan L Littlehale[1], John F Marko[2,3], Michael T Laub[1,4]\***

[1]Department of Biology, Massachusetts Institute of Technology, Cambridge, United States; [2]Department of Molecular Biosciences, Northwestern University, Evanston, United States; [3]Department of Physics and Astronomy, Northwestern University, Evanston, United States; [4]Howard Hughes Medical Institute, Massachusetts Institute of Technology, Cambridge, United States

**Abstract** Supercoiling impacts DNA replication, transcription, protein binding to DNA, and the three-dimensional organization of chromosomes. However, there are currently no methods to directly interrogate or map positive supercoils, so their distribution in genomes remains unknown. Here, we describe a method, GapR-seq, based on the chromatin immunoprecipitation of GapR, a bacterial protein that preferentially recognizes overtwisted DNA, for generating high-resolution maps of positive supercoiling. Applying this method to *Escherichia coli* and *Saccharomyces cerevisiae*, we find that positive supercoiling is widespread, associated with transcription, and particularly enriched between convergently oriented genes, consistent with the 'twin-domain' model of supercoiling. In yeast, we also find positive supercoils associated with centromeres, cohesin-binding sites, autonomously replicating sites, and the borders of R-loops (DNA-RNA hybrids). Our results suggest that GapR-seq is a powerful approach, likely applicable in any organism, to investigate aspects of chromosome structure and organization not accessible by Hi-C or other existing methods.

**\*For correspondence:**
msguo@uw.edu (MSG);
laub@mit.edu (MTL)

**Present address:** [†]Department of Microbiology, University of Washington School of Medicine, Seattle, United States

## Introduction

The DNA inside every cell can adopt a wide range of topologies. Genomic DNA can become supercoiled when the DNA duplex winds about its own axis. For plectonemic DNA, this supercoiling can manifest as writhe, with the DNA forming a left-handed superhelix (positive supercoiling) or a right-handed superhelix (negative supercoiling). As DNA writhe can interconvert with twist, positive and negative supercoils can also manifest as over- or undertwisted DNA, respectively. Because overtwisted DNA inhibits strand melting and undertwisted DNA promotes it, DNA supercoiling can profoundly impact the binding of regulatory proteins, promoter firing dynamics, DNA replication, and chromosome architecture (*Dillon and Dorman, 2010*; *Gilbert and Allan, 2014*). Despite the importance of DNA topology, the location and distribution of supercoils in genomes remain virtually unknown.

Supercoils are introduced by the translocation of RNA polymerase. When the DNA duplex is unwound during transcription, positive supercoils occur ahead of the polymerase and negative supercoils in its wake, producing the 'twin-domain' model of supercoiling (*Liu and Wang, 1987*; *Wu et al., 1988*). Supercoils can then diffuse into neighboring loci, though how far they travel and what factors restrict their movement are not well understood (*Gilbert and Allan, 2014*). Supercoils can also be introduced and removed by DNA topoisomerases, enzymes that transiently break and rejoin the DNA backbone (*Pommier et al., 2016*; *Vos et al., 2011*). Topoisomerase activity is essential for DNA replication, with the rapid removal of the positive supercoils ahead of the replication fork necessary to prevent replisome arrest (*Postow et al., 2001*). The extent to which supercoils are

persistent in genomes or rapidly removed by topoisomerases is not clear. Our understanding of how supercoiling impacts chromosome organization and function is severely limited by a lack of high-resolution methods for mapping supercoils in living cells and the inability to specifically interrogate positive supercoiling.

Chromosome conformation capture technologies such as Hi-C have dramatically altered our understanding of chromosome organization. However, Hi-C typically has a resolution of only 5–10 kb and does not capture supercoiling, which generally operates on shorter length scales (*Kempfer and Pombo, 2020*). Classic methods to interrogate supercoiling, for example, ultracentrifugation of whole chromosomes or plasmid electrophoresis, only infer average supercoiling, and other methods, which rely on supercoiling-dependent promoters or recombination frequencies, have limited throughput, precluding genome-scale studies (*Corless and Gilbert, 2017*; *Higgins, 2017*). More recently, supercoiling has been measured via preferential crosslinking of psoralen derivatives to undertwisted, negatively supercoiled DNA (*Achar et al., 2020*; *Bermúdez et al., 2010*; *Kouzine et al., 2013*; *Lal et al., 2016*; *Naughton et al., 2013*; *Sinden et al., 1980*; *Teves and Henikoff, 2014*). Consequently, psoralen-based studies can infer the presence of positive supercoiling at regions with decreased crosslinking. However, RNA polymerase, nucleosomes, DNA-binding proteins, or unwound DNA could each block psoralen intercalation and complicate the interpretation of crosslinking efficiency (*Bermúdez et al., 2010*; *Toussaint et al., 2005*; *Wellinger and Sogo, 1998*). These issues could impact the conclusions from a psoralen-based study suggesting that coding regions in yeast are positively supercoiled, with negatively supercoiled DNA accumulating at gene boundaries (*Achar et al., 2020*), a finding in apparent conflict with the twin-domain model of supercoiling.

Here, we develop a high-resolution method to probe the distribution of positive supercoils in cells. Our approach, GapR-seq, is based on chromatin immunoprecipitation (ChIP) sequencing of GapR, a bacterial protein that preferentially binds overtwisted DNA. Our previous work in the bacterium *Caulobacter crescentus* demonstrated that GapR localizes to the 3' ends of highly transcribed regions and is required, together with type II topoisomerases, to relax positively supercoiled DNA during replication (*Guo et al., 2018*). We showed with in vitro topological assays and a crystal structure that GapR likely binds overtwisted DNA (*Guo et al., 2018*). We now show, using single-molecule magnetic tweezer (MT) experiments, that GapR preferentially recognizes positively supercoiled DNA and has less affinity for negatively supercoiled DNA. These results suggested that GapR could serve as a sensor of positive supercoils in any cell, which we tested in *Escherichia coli* and *Saccharomyces cerevisiae*. In both organisms, GapR-seq yields strong signal in intergenic regions known or expected to harbor positively supercoiled DNA, accumulating downstream of highly transcribed regions, particularly between convergently oriented genes. This provides an important check for applicability in eukaryotic chromatin, which has been observed to have low torsional stiffness to positive torsional stress (*Le et al., 2019*). In yeast, we also find positively supercoiled DNA associated with centromeres, cohesin-binding sites, and autonomously replicating sequences. GapR-seq further suggests that overtwisted DNA may be associated with the boundaries of DNA-RNA hybrids, or R-loops. Thus, taken together our work demonstrates that GapR-seq is a powerful new approach for mapping positive supercoils and investigating how they shape the structure and function of chromosomes in all kingdoms of life.

## Results

### GapR interacts with overtwisted, positively supercoiled DNA

We previously showed that GapR binds at sites of expected positive supercoiling in *Caulobacter* cells and that purified GapR binds to overtwisted DNA in vitro (*Guo et al., 2018*), suggesting that GapR could be used as a probe for positive supercoiling. We first validated our prior in vitro findings by performing a topological assay in which a circular, nicked plasmid was incubated with GapR and then treated with T4 DNA ligase to trap any supercoils constrained by GapR. After protein removal by Proteinase K, the resulting changes in plasmid topology will reflect the topological binding preference of GapR. Increasing amounts of GapR led to a gradual, but marked change in plasmid topology (*Figure 1A*, *Figure 1—figure supplement 1A*), leading to the formation of positively

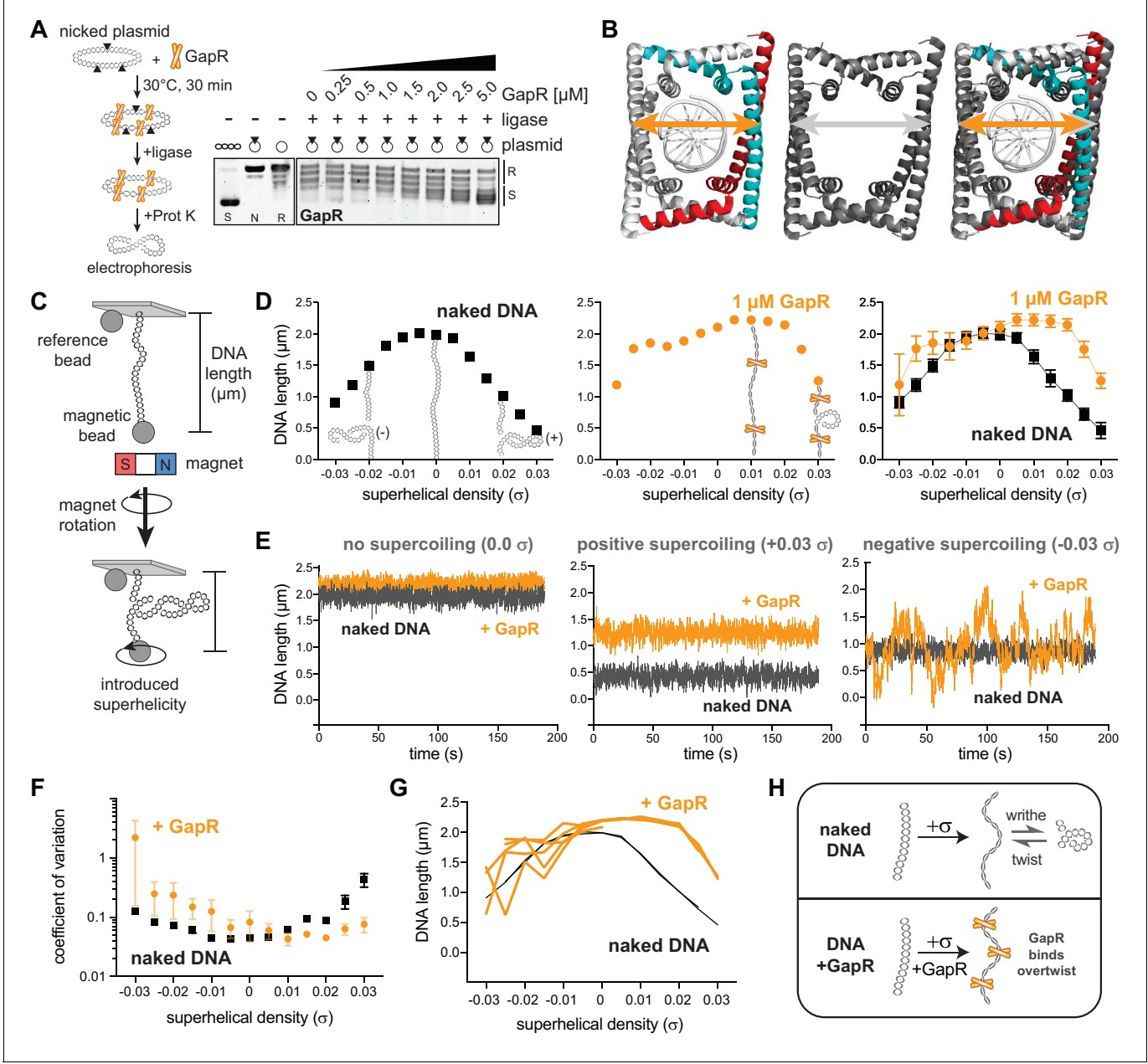

**Figure 1.** GapR interacts stably with overtwisted, positively supercoiled DNA. (A) GapR DNA topology assay. GapR was incubated with nicked plasmid before treatment with T4 DNA ligase and subsequent quenching, deproteinization, and electrophoresis (schematic). Gel analysis of plasmid topology with positively supercoiled (S), nicked (N), and relaxed (R) standards. (B) Comparison of GapR-DNA crystal structures. Left, 6GC8 (*Guo et al., 2018*); middle, 6OZX (*Tarry et al., 2019*); right, overlay. Diameter of 6GC8 (orange arrow) and 6OZX (gray arrow) indicated. (C) Schematic of magnetic tweezer (MT) experiment. See also *Figure 1—figure supplement 1C*. (D) Behavior of naked DNA (left), DNA incubated with 1 µM GapR (middle), and overlay (right) in a rotation-extension experiment with the corresponding DNA conformation superimposed. Data indicate mean ± SD, n = 200 at each σ, in a single MT experiment. (E) DNA ± 1 µM GapR behavior over time from D under no supercoiling (σ = 0.0, left), positive supercoiling (σ = +0.03, middle), and negative supercoiling (σ = −0.03, right). (F) Coefficient of variation of force-extension experiments of DNA ± 1 µM GapR. Data indicate mean ± SEM, n ≥ 3. (G) Hysteresis of force-extension experiments. Traces indicate multiple rotation-extension measurements from one DNA molecule ± 1 µM GapR. (H) Model of GapR binding to overtwisted DNA.

The online version of this article includes the following source data and figure supplement(s) for figure 1:

**Source data 1.** Raw gels associated with *Figure 1A*.

**Figure supplement 1.** GapR binding to supercoiled DNA in a magnetic tweezer (MT) experiment.

**Figure supplement 1—source data 1.** Raw gels associated with *Figure 1—figure supplement 1A*.

**Figure supplement 1—source data 2.** Raw gels associated with *Figure 1—figure supplement 1B*.

supercoiled plasmid as determined using two-dimensional chloroquine electrophoresis (*Figure 1—figure supplement 1B*).

In addition to these topological assays, our previous crystal structure (*Guo et al., 2018*) captured GapR as a dimer-of-dimers that fully encircled DNA, without any base-specific contacts and with a narrow DNA-binding cavity that should preferentially accommodate overtwisted DNA (*Figure 1B*). Subsequently, other crystal structures of GapR in complex with DNA were solved (*Huang et al., 2020*; *Lourenço et al., 2020*; *Tarry et al., 2019*) and featured a larger GapR cavity (*Figure 1B*), leading to a proposal that GapR does not have a topological preference for DNA. However, crystal structures cannot reveal whether GapR preferentially binds supercoiled DNA. Therefore, we turned to MTs to interrogate GapR binding to single DNA molecules with controlled superhelical density (σ).

Briefly, one end of an 11.4 kb dsDNA fragment was immobilized to the coverslip of a flow cell while the other end was bound to a magnetic bead (*Figure 1C*, *Figure 1—figure supplement 1C*). The flow cell was then placed on top of a magnet, so that rotation of the magnet introduces over- or undertwisting of the DNA; at low forces (~0.3 pN), the DNA then adopts either positive or negative supercoiling (writhe), which shortens the DNA molecule. This structural change is observed by measuring DNA length, i.e. the distance between the magnetic bead and a reference bead fixed to the coverslip (*Figure 1C*), using quantitative microscopy.

We first characterized the behavior of naked DNA by measuring its length at various σ (from σ = –0.03 to +0.03 σ), generating an extension versus rotation or 'hat' curve (*Figure 1D*, *Figure 1—figure supplement 1D*). Acquisition of data in accord with prior studies (*Bai et al., 2011*; *Sun et al., 2013*) is important in that it validates that the bead is tethered by one DNA of the expected molecular length. We note that the naked DNA hat curves for DNA tension of 0.3 pN (*Figure 1D*, left) extrapolate to zero extension for negative supercoiling density of approximately σ = –0.05, indicating that the plectonemes being formed in naked DNA under this tension have torque and writhing density of this value (*Marko, 2007*), close to the level of supercoiling of DNA found in *E. coli* (*Higgins, 2016*). The MT data reported in this paper were all acquired at DNA tension of 0.3 pN for this reason.

Following single-DNA validation, we then added GapR (at 10, 100, or 1000 nM) to the relaxed DNA and repeated the rotation-extension measurement (*Figure 1D*, *Figure 1—figure supplement 1D–F*). We note that this concentration range corresponds to that encountered by DNA in *Caulobacter*, where 2000–3000 copies of the protein are found in a cell of cytoplasmic volume of approximately 2 µm$^3$ (recall 1 nM ≈ 0.6 molecules per µm$^3$) (*Guo et al., 2018*). After introducing positive σ, we observed significantly increased extension of GapR-bound DNA compared to naked DNA (*Figure 1D*, *Figure 1—figure supplement 1D–F*). These results indicate that GapR constrains the added positive σ, preventing writhing, and increasing DNA extension (*Figure 1D*, *Figure 1—figure supplement 1D–F*). At 1 µM GapR, DNA extension was longest at +0.015 σ. Further increasing σ reduces DNA extension because the additional positive σ cannot be constrained by GapR and converts to writhe (*Figure 1D*, *Figure 1—figure supplement 1D*). The shift of the hat curve peaks to positive σ is expected for a protein that has a higher affinity for overtwisted versus undertwisted DNA, that is that overtwists DNA upon binding (*Yan and Marko, 2003*).

We did not observe any tendency of GapR to reduce DNA extension near the peak of the hat curves, as would occur if it introduced appreciable DNA bending, chiral coiling, or DNA crossbridging, as can be observed for other types of proteins (*Skoko et al., 2004*; *Sun et al., 2013*). Instead, GapR slightly increases overall DNA extension at the peak of the hat curves (*Figure 1D*, *Figure 1—figure supplement 1D–F*), possibly due to stretching of double helix secondary structure, or modification of double helix effective persistence length (*Yan and Marko, 2003*). These MT data, together with our in vitro topological assays (*Guo et al., 2018*), support a model that GapR binds overtwisted DNA.

We observed that the experiment-to-experiment variability of mean extension of 1000 nM GapR-DNA was considerably larger at negative σ than at positive σ or compared to undertwisted naked DNA (*Figure 1D*, *Figure 1—figure supplement 1D, G*). Moreover, in individual experiments, the length of GapR-bound DNA molecules dynamically fluctuated at negative σ as a function of time, leading to a larger standard deviation of extension at negative σ than for positive σ (*Figure 1E*, *Figure 1—figure supplement 1G*), and with a substantially larger coefficient of variation in DNA length at negative σ compared to positive σ or naked DNA (*Figure 1F*). Therefore, the structures of GapR-

DNA complexes at negative σ are less stable than those at positive σ. These behaviors were reversible and did not display hysteresis; we performed multiple rotation-extension experiments on the same GapR-bound DNA, finding that GapR-DNA stably maintained its length when overtwisted, but varied in length substantially when undertwisted (*Figure 1G*). To our knowledge, these behaviors are unique to GapR. MT studies of other DNA-binding proteins have not reported analogous supercoiling-dependent instability in DNA length (*Ding et al., 2014*; *Sun et al., 2013*; *Vlijm et al., 2017*; *Zorman et al., 2012*).

Given the strong variation in DNA length resulting from changing linking number from negative to positive and back to positive (*Figure 1G*) and the large dynamical variation in DNA length (*Figure 1E*, right), we carried out experiments where we first prepared GapR-DNA complexes at 1000 nM GapR, and then replaced the flow cell contents with reaction buffer lacking GapR, thus 'washing' the protein in solution away. We found that the hat curves (*Figure 1—figure supplement 1G*) and the strong dynamical fluctuations for negative supercoiling (*Figure 1—figure supplement 1G*) persisted for more than 30 min post-wash in the absence of GapR in solution (*Figure 1—figure supplement 1H, I*). Given that this persistence time is far longer than the ≈30 s timescale for extension variations (*Figure 1E*, *Figure 1—figure supplement 1G*), complete dissociation of GapR from DNA is not a viable explanation of the variability and dynamics of extension for negative supercoiling in the presence of GapR in solution (*Figure 1E*). We propose that GapR may rapidly diffuse along or perhaps partially dissociate from negatively supercoiled DNA, and that the organization of GapR-DNA complexes for negative supercoiling is unstable, possibly due to a combination of GapR sliding and hopping with dynamic reorganization of DNA supercoiling. Whatever the case, GapR stably interacts with positively supercoiled DNA (*Figure 1H*), indicating that GapR could be used as a positive supercoil sensor.

## GapR is associated with positive supercoils in *Escherichia coli*

To test if GapR could be used to monitor positive supercoiling in cells, we placed GapR-3xFLAG under tetracycline-inducible control in *E. coli*, an organism without a GapR homolog, and performed chromatin immunoprecipitation-sequencing (ChIP-seq) after inducing GapR (*Figure 2—figure supplement 1A*). Importantly, GapR induction did not affect the growth rate of *E. coli*, alter global transcription, or the expression of known supercoiling-sensitive genes (*Peter et al., 2004*; *Figure 2—figure supplement 1B–D*). Comparing the ChIP of GapR-3xFLAG to an untagged GapR control revealed hundreds of reproducible peaks throughout the *E. coli* chromosome (*Figure 2—figure supplement 1A, E*). As in *Caulobacter* (*Guo et al., 2018*), we found a modest correlation between GapR binding and AT-rich DNA, but AT-content alone cannot explain or predict the distribution of GapR (*Figure 2—figure supplement 1F*).

Because positive supercoils are introduced into DNA by RNA polymerase (*Liu and Wang, 1987*; *Wu et al., 1988*), they should localize within or downstream of highly expressed genes and transcription units (TUs; *Figure 2A*). We therefore compared our GapR ChIP and RNA-seq profiles. At a highly expressed ribosomal protein operon (*Figure 2B*), we observed GapR binding from just inside the 3' end of *rplQ* to ~2 kb downstream (see also *Figure 2—figure supplement 1G*). To test if this GapR binding was transcription-dependent, we treated cells with the RNA polymerase inhibitor rifampicin for 20 min before performing GapR ChIP. Consistent with our results in *Caulobacter* (*Guo et al., 2018*), rifampicin largely abrogated GapR binding downstream of *rplQ* (*Figure 2B*, see also *Figure 2—figure supplement 1G*).

Next, we asked if GapR was enriched within genes or at the 5' and 3' ends of TUs (i.e., genes or operons). Strikingly, at *rplQ* and at other highly expressed TUs (*Figure 2B*, *Figure 2—figure supplement 1G*), GapR bound at the 3' end of transcripts and was largely unenriched within TUs (*Figure 2—figure supplement 1H*). These findings are consistent with the predictions of the 'twin-domain' model (*Liu and Wang, 1987*; *Wu et al., 1988*). If TUs are covered by multiple, closely spaced RNA polymerases, positive supercoils introduced ahead of one RNA polymerase will be eliminated by negative supercoils that arise in the wake of the downstream polymerase. Consequently, the positive supercoiling associated with transcription is predicted to accumulate at the 3' ends of transcribed genes (*Figure 2C*), as observed.

To quantitatively assess how GapR binding is associated with positive supercoiling, we compared GapR binding at the 5' and 3' ends of all long (≥ 1500 bp) TUs (i.e., genes or operons), normalized in each case to enrichment within the TU, observing significant occupancy of GapR only at the 3'

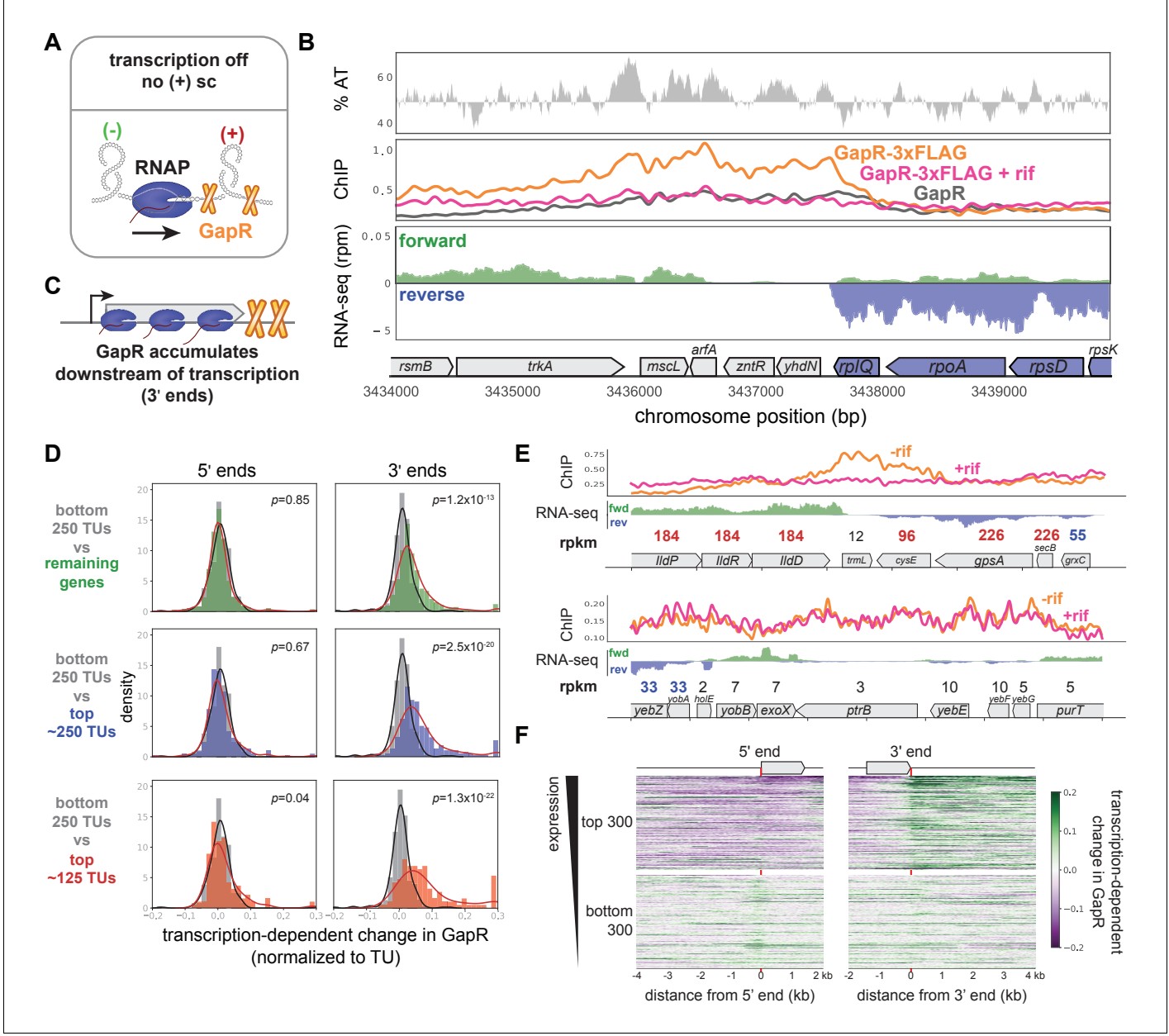

**Figure 2.** GapR is associated with positive supercoiling in *E. coli*. (**A**) Positive supercoiling is generated downstream of RNA polymerase during transcription as predicted by the 'twin-domain' model. (**B**) GapR chromatin immunoprecipitation (ChIP) profiles at a highly expressed operon. AT content (top), with AT content below the genomic average (50%) plotted in reverse. ChIP-seq (middle) of untreated (orange) or rifampicin-treated (pink) GapR-3xFLAG cells and untreated GapR cells (gray). Transcription from the forward (green) and reverse (blue) strands with the position of annotated genes indicated (bottom). (**C**) GapR and positive supercoiling accumulates at the 3' end of genes, not within genes. (**D**) Transcription-dependent change in GapR ChIP at 5' (left) or 3' (right) ends normalized by binding within the transcription unit (TU) at different expression thresholds. Student's t-test p-value shown. (**E**) Examples of GapR-3xFLAG ChIP without (orange) or with (pink) rifampicin treatment. Transcription of the forward (green) and reverse (blue) strands with annotated genes indicated. Expression values are colored using the same rpkm cutoffs as in **D**. (**F**) Heatmap showing transcription-dependent change in GapR around 5' and 3' ends for the top and bottom 300 long TUs sorted by expression.

The online version of this article includes the following figure supplement(s) for figure 2:

**Figure supplement 1.** *E. coli* GapR chromatin immunoprecipitation-sequencing (ChIP-seq).

**Figure supplement 2.** GapR full-length chromatin immunoprecipitation-sequencing (ChIP-seq).

ends (t-test, p<10$^{-10}$, *Figure 2—figure supplement 1I*). To determine if GapR binding is transcription-dependent, we calculated the change in GapR enrichment within and near the 5′ and 3′ ends of all long TUs following rifampicin treatment. The distribution of changes at 5′ ends was symmetric and centered around 0, with the distribution of changes for the 3′ ends significantly shifted to the right (t-test, p<10$^{-13}$, *Figure 2—figure supplement 1J*), indicating that GapR binding near the 3′ ends of TUs is sensitive to transcription.

If GapR is recognizing transcription-dependent positive supercoiling, binding should correlate with transcriptional strength. To test this idea, we compared GapR ChIP and transcription-dependent GapR enrichment at long TUs at various expression levels (*Figure 2D, E*, *Figure 2—figure supplement 1K*, see Materials and methods). GapR binding at the 5′ end relative to within the TU was not dependent on expression level (t-test, p>0.01 for all expression cutoffs, *Figure 2D*, *Figure 2—figure supplement 1K*). In contrast, at 3′ ends, GapR binding was dependent on expression, with highly expressed TUs having significantly increased GapR occupancy relative to within the TU (t-test, p<10$^{-13}$ for all expression cutoffs, *Figure 2D, E*, *Figure 2—figure supplement 1K*). We also ordered long TUs by expression level and plotted as a heatmap the transcription-dependent change in GapR surrounding the 5′ and 3′ ends. These heatmaps clearly demonstrated that GapR was enriched specifically after the termination site of highly expressed TUs, with GapR occupancy typically extending several kb downstream (*Figure 2F*, *Figure 2—figure supplement 1L*). In contrast, GapR binding was de-enriched at the 5′ ends of and within highly expressed genes. Notably, GapR was not found at the 3′ ends of all well-expressed TUs. However, when we examined exceptions further, we found that these TUs were oriented in tandem with other highly expressed genes, such that GapR accumulated at the 3′ end of the downstream TU (*Figure 2—figure supplement 2A*). Likewise, GapR enrichment at the 3′ ends of poorly expressed genes (or at 5′ ends) was typically attributable to the effects of a well-expressed TU on the opposite strand (*Figure 2—figure supplement 2B*). Collectively, these analyses support the conclusion that GapR is localized to the positive supercoils produced by transcription in *E. coli*.

## GapR recognizes positive supercoiling as a tetramer

While striking, our ChIP results cannot exclude the possibility that GapR is localized downstream of transcription simply because such DNA is more accessible. To control for this possibility, we sought GapR mutants that bound DNA but no longer recognize DNA topology. Previous work demonstrated that truncations in the C-terminal tetramerization domain generated constitutively dimeric GapR (GapR$^{1-76}$) (*Huang et al., 2020*; *Lourenço et al., 2020*). Because this dimeric GapR cannot encircle the DNA duplex yet retains all of the DNA binding residues of GapR, we reasoned that this variant would bind DNA without recognizing supercoiling.

To test this hypothesis, we expressed and purified dimeric GapR$^{1-76}$ (*Figure 3—figure supplement 1A*). Electrophoretic mobility shift assays (EMSA) showed that GapR$^{1-76}$ binds DNA, albeit with lower affinity than full-length GapR (*Figure 3—figure supplement 1B*). We then asked if GapR$^{1-76}$ binds positive supercoiling by performing topological assays comparing the supercoiling preference of GapR and GapR$^{1-76}$. Whereas full-length GapR trapped positive supercoils, GapR$^{1-76}$ did not alter plasmid topology (*Figure 3A*, *Figure 3—figure supplement 1C*, see also *Figure 1A*, *Figure 1—figure supplement 1A, B*). We conclude that dimeric GapR$^{1-76}$ binds DNA but no longer recognizes DNA topology, indicating that positive supercoiling recognition requires a tetrameric conformation.

To validate that tetrameric GapR is recognizing positive supercoiling in vivo, we compared the ChIP profiles of full-length GapR-3xFLAG with GapR$^{1-76}$-3xFLAG. Notably, GapR$^{1-76}$ does not bind at the 3′ ends of TUs as seen with full-length GapR (*Figure 3B*, *Figure 3—figure supplement 1D*). In fact, the ChIP profiles of GapR$^{1-76}$ and full-length GapR were not correlated (*Figure 3C*), demonstrating that full-length GapR is not simply bound to accessible DNA. Altogether, our data strongly support the idea that GapR is recognizing overtwisted, positive supercoiled DNA in vivo. We propose that GapR-seq provides a direct, high-resolution readout of positive supercoiling in living cells.

## Positive supercoils accumulate in regions of convergent transcription

Because positive supercoils are generated downstream of translocating RNA polymerase, we hypothesized that these supercoils, and GapR, should be strongly associated with convergently oriented TUs (*Figure 3D*). Indeed, we found that GapR, but not GapR$^{1-76}$, was

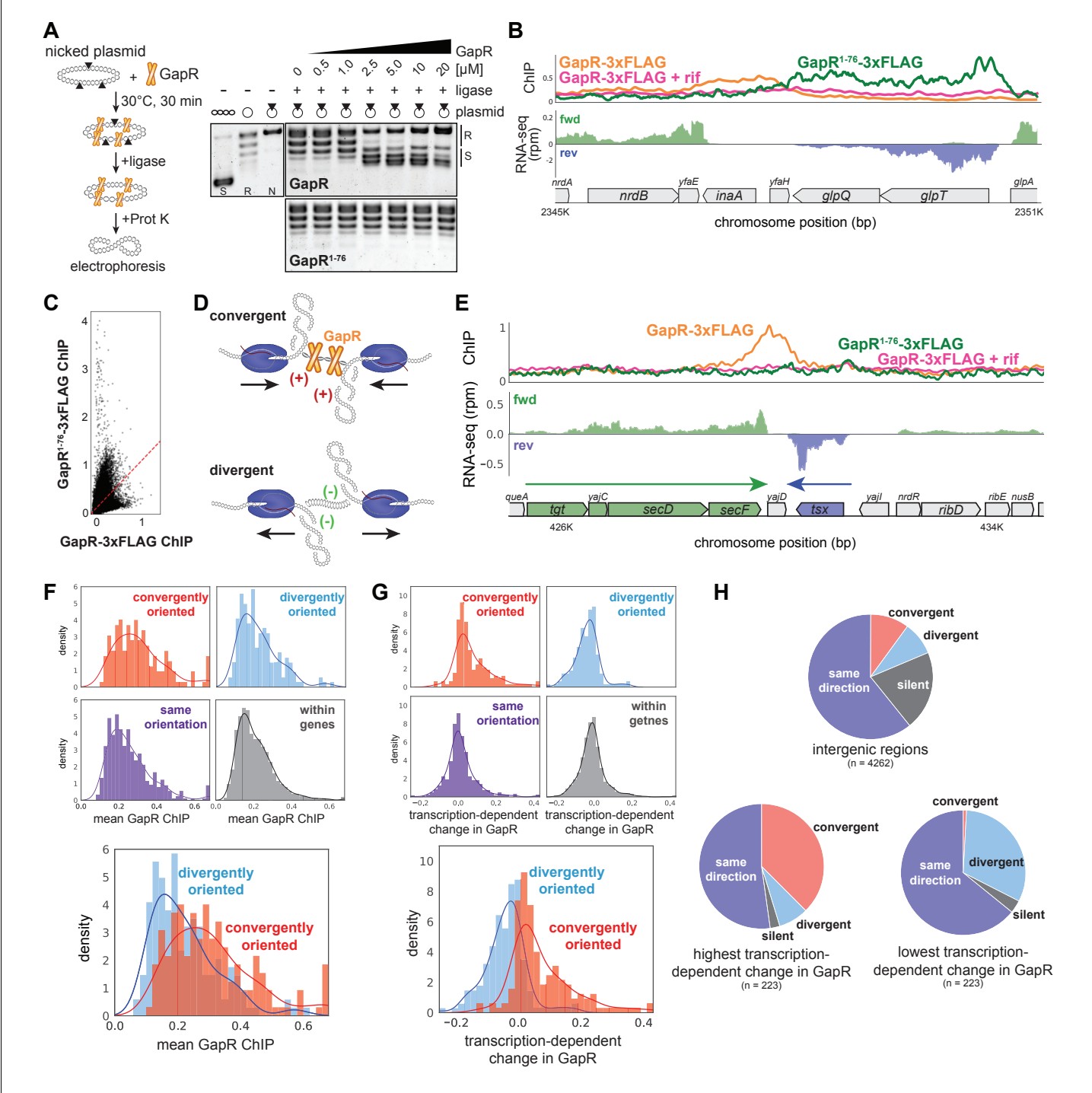

**Figure 3.** GapR recognizes DNA supercoiling and is associated with convergent transcription. (**A**) GapR[1-76] does not recognize DNA topology. Full-length GapR and GapR[1-76] were incubated with nicked plasmid before treatment with T4 DNA ligase and subsequent quenching, deproteinization, and electrophoresis (schematic). Gel analysis of plasmid topology with supercoiled and relaxed standards as in *Figure 1A*. (**B**) GapR[1-76]-3xFLAG chromatin immunoprecipitation (ChIP) (green) and GapR-3xFLAG ChIP without (orange) and with (pink) rifampicin treatment (top). Transcription of the forward (green) and reverse (blue) strands with annotated genes indicated. (**C**) Correlation between GapR-3xFLAG and GapR[1-76]-3xFLAG ChIP experiments. (**D**) Positive supercoils are trapped by convergent transcription. (**E**) ChIP of GapR[1-76]-3xFLAG (green) and GapR-3xFLAG without (orange) and with (pink) rifampicin treatment at convergently oriented transcription units (TUs). Transcription of the forward (green) and reverse (blue) strands with annotated genes indicated. (**F**) GapR ChIP in gene bodies (dark gray), in divergent regions (blue), convergent regions (red), and where transcription is in the same orientation (purple). Overlay of divergent and convergent regions (bottom). (**G**) Transcription-dependent changes in GapR plotted as in (**F**). (**H**) Regions

*Figure 3 continued on next page*

*Figure 3 continued*

with high transcription-dependent change in GapR are more frequently between convergent genes. Pie charts summarize the orientation of flanking genes for all intergenic regions (top) and intergenic regions with highest (bottom left) or lowest (bottom right) transcription-dependent change in GapR.

The online version of this article includes the following source data and figure supplement(s) for figure 3:

**Source data 1.** Raw gels associated with *Figure 3A*.
**Figure supplement 1.** GapR full-length and truncation variant chromatin immunoprecipitation-sequencing (ChIP-seq).
**Figure supplement 1—source data 1.** Raw gels associated with *Figure 3—figure supplement 1A*.
**Figure supplement 1—source data 2.** Raw gels associated with *Figure 3—figure supplement 1B*.
**Figure supplement 1—source data 3.** Raw gels associated with *Figure 3—figure supplement 1C*.

frequently enriched between convergently oriented operons in *E. coli* (*Figure 3E, F*). Convergently oriented regions had higher GapR signal compared to intragenic or divergently oriented regions (t-test, $p < 10^{-9}$, *Figure 3F*, examples in *Figure 3—figure supplement 1F, G*), whereas GapR$^{1-76}$ bound similarly in convergently and divergently oriented regions (*Figure 3—figure supplement 1E*). Importantly, GapR enrichment between convergently oriented TUs was dependent on transcription (t-test, $p < 10^{-27}$; *Figure 3G*).

To further validate the association between GapR, positive supercoiling, and convergently oriented TUs, we selected the ~220 genomic regions showing the highest and lowest transcription-dependent changes in GapR ChIP (see Materials and methods) and asked how TUs were oriented around these regions. Regions with the highest transcription-dependent changes in GapR were highly enriched for convergently oriented TUs compared to regions with the lowest transcription-dependence or intergenic regions (Fisher's exact test, $p < 10^{-35}$ and $p < 10^{-14}$, respectively, *Figure 3H*). Together, our in vitro and in vivo data demonstrate that GapR ChIP effectively reads out the locations of overtwisted, positive supercoiled DNA in living cells. Furthermore, our results validate the 'twin-domain' model of supercoiling and reveal that persistent positive supercoils arise downstream of active TUs and are trapped by converging RNA polymerases in bacterial cells.

## GapR is associated with positive supercoiling in *S. cerevisiae*

Next, we asked if our GapR-seq method could be extended to the budding yeast, *S. cerevisiae*, which also does not encode a GapR homolog. We integrated either *gapR* or *gapR-3xFLAG* into *S. cerevisiae* at *LEU2* under control of the *GAL1-10* promoter. We grew cells to exponential phase in the presence of raffinose to repress GapR expression, and then induced GapR for 6 hr with galactose before performing ChIP-seq. As in bacteria, (1) expression of GapR-3xFLAG did not significantly alter the transcriptional profile of yeast (*Figure 4—figure supplement 1A*), with less than twofold changes in genes that are supercoiling-sensitive (*Pedersen et al., 2012*) or involved in the general stress response (Pka1, Hog1, Hsf1, Yap1), the unfolded protein response, or the DNA damage response (*Jaehnig et al., 2013*; *Figure 4—figure supplement 1B*); (2) GapR-3xFLAG was reproducibly enriched at specific sites in the genome when compared to the untagged control (*Figure 4—figure supplement 1C, D*); and (3) there was only modest correlation between GapR ChIP and local AT-content (*Figure 4—figure supplement 1E*).

As in bacteria, we frequently observed GapR peaks at, and extending beyond, the 3' ends of most genes, with peaks almost never occurring within coding regions, and extending ~900 bp long on average, somewhat shorter than seen in *E. coli* (*Figure 4A*, *Figure 4—figure supplement 1F, G*). Also as in bacteria, GapR was significantly enriched at the 3', but not the 5', ends of genes (*Figure 4B*). To determine if GapR is recognizing transcription-dependent positive supercoiling, we computationally compared our GapR-seq and RNA-seq profiles. We found that GapR enrichment at the 3' ends of genes was clearly correlated with transcriptional strength (t-test, $p < 10^{-25}$ for all expression cutoffs at 3' ends, *Figure 4C, D*). Additionally, and again as found in *E. coli*, GapR was enriched at regions of convergent transcription compared to divergent or intragenic regions (*Figure 4E, F*, see Materials and methods). We observed identical results when we performed an analogous experiment for cells grown in glycerol before GapR induction (*Figure 4—figure supplement 1G–J*).

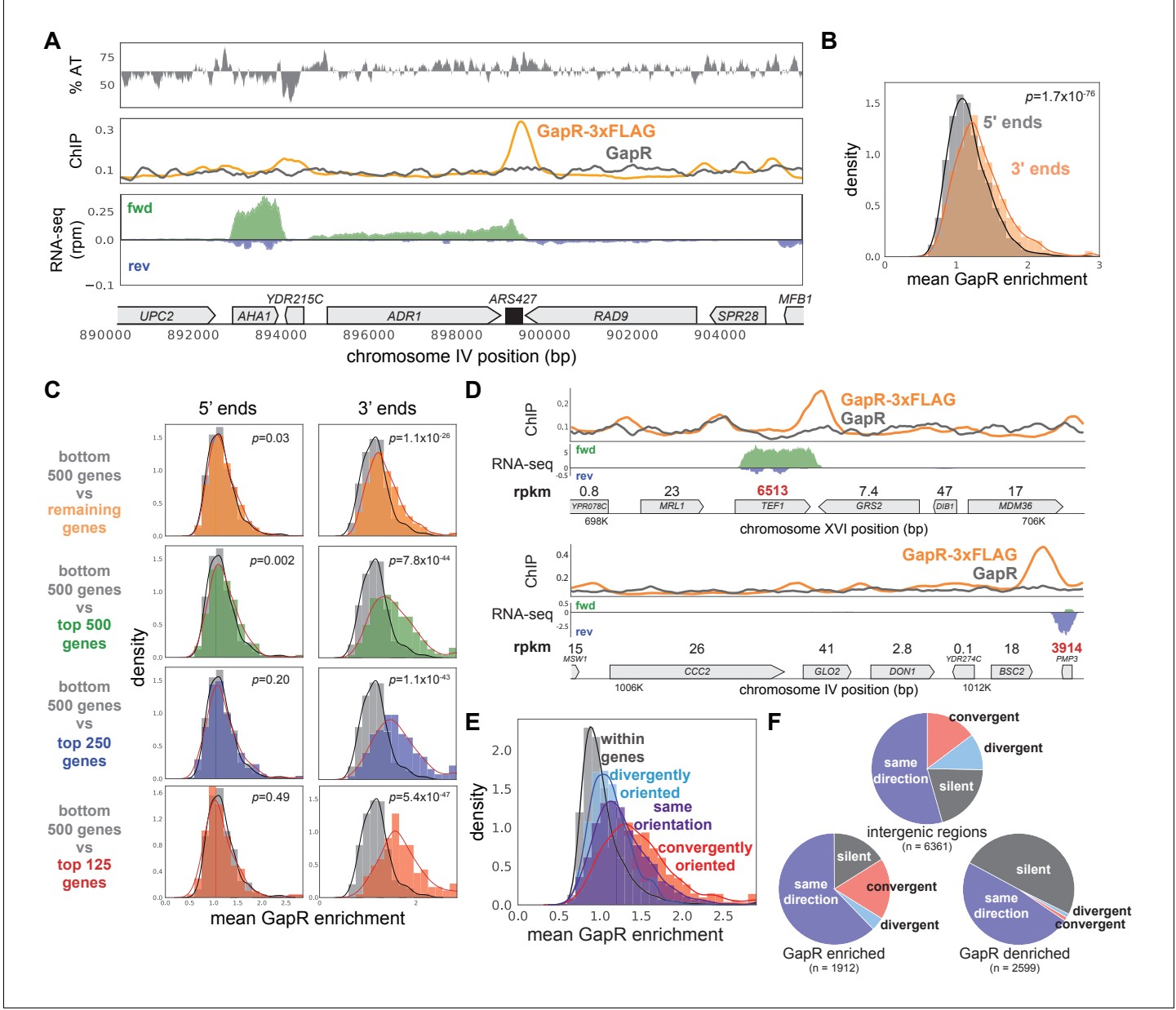

**Figure 4.** GapR is associated with positive supercoiling in yeast. (**A**) Chromatin immunoprecipitation (ChIP) of *S. cerevisiae* grown in raffinose before GapR induction. AT content (top), ChIP-seq (middle) of GapR-3xFLAG (orange) or untagged GapR (gray) expressing cells. Transcription of the forward (green) and reverse (blue) strands with annotated genes indicated (bottom). (**B**) Mean GapR enrichment (GapR-3xFLAG ChIP normalized by untagged ChIP) in a 500 bp window at the 5′ and 3′ end of long genes. Student's t-test p-value is shown. (**C**) Mean GapR enrichment at 5′ and 3′ ends of long genes at various transcriptional cutoffs. Student's t-test p-value is shown. (**D**) Examples of GapR-3xFLAG (orange) and untagged GapR (gray) ChIP. Transcription of the forward (green) and reverse (blue) strands with annotated genes indicated. Expression values are colored using the same rpkm cutoffs as in (**C**). (**E**) GapR is enriched between convergently oriented genes. Student's t-test, convergent versus all other regions, $p<10^{-56}$. (**F**) GapR-bound regions are more frequently between convergent genes. Fisher's exact test, GapR-enriched versus -denriched, $p<10^{-13}$. Pie charts shown as in *Figure 3H*.

The online version of this article includes the following figure supplement(s) for figure 4:

**Figure supplement 1.** *S. cerevisiae* GapR chromatin immunoprecipitation-sequencing (ChIP-seq).

Because GapR was enriched within the 5' ends of many poorly expressed genes, we asked if this enrichment can be explained by positive supercoiling generated from an upstream transcript. Indeed, we found that between co-oriented genes, GapR was correlated with the transcriptional level of the upstream gene but not the downstream gene (*Figure 4—figure supplement 1K*). Restricting this analysis to gene pairs in which the downstream gene is poorly expressed also showed that GapR occupancy only correlated with the transcription level of the upstream gene (*Figure 4—figure supplement 1K*). Taken altogether, our results suggest that GapR-seq identifies *S. cerevisiae* genomic regions harboring positive supercoils, and that this topology typically arises downstream of highly expressed genes and particularly between convergently oriented genes. These observations of GapR sensitivity at expected loci of positive torsional stress are key validations of our approach in eukaryotic chromatin, which has been observed to have low stiffness to positive torsional stress (*Le et al., 2019*).

## GapR binding in *S. cerevisiae* is responsive to transcription

To further validate that GapR is recognizing transcription-dependent positive supercoiling, we arrested cells in G1 for 2 hr with α-factor before inducing GapR. Compared to cycling cells, α-factor arrested cells upregulate genes required for mating and downregulate genes specific to S and M phases (*Figure 5—figure supplement 1A*). Thus, upon α-factor arrest, we anticipated increased GapR enrichment at the 3' ends of upregulated genes and decreased GapR occupancy at the 3' ends of downregulated genes. Indeed, some of the largest changes in GapR-seq arose near genes known to be induced or repressed during mating such as *FIG1* or *YGP1* (*Figure 5A*, *Figure 5—figure supplement 1B*).

To quantitatively assess how GapR binding is affected by altered transcription, we first examined α-factor-dependent GapR binding at the 5' and 3' ends of each gene. As anticipated, GapR occupancy increased at the 3' ends of genes induced in α-factor and modestly decreased at the 3' ends of genes repressed by α-factor (*Figure 5—figure supplement 1C*). To better visualize these changes, we ordered genes by their change in expression in α-factor and plotted as a heatmap the change in GapR at each gene's 5' and 3' ends (*Figure 5B*). These heatmaps showed that upon α-factor treatment GapR binding was often substantially increased at the 3' ends of upregulated genes and decreased at the 3' ends of downregulated genes (*Figure 5B*). Collectively, these results demonstrate that GapR-seq reveals transcription-dependent positive supercoiling in *S. cerevisiae*, as it does in *E. coli* and *C. crescentus*. Further, our data validate the 'twin-domain model' in *S. cerevisiae*, revealing that persistent positive supercoils are found downstream of actively transcribed genes.

## GapR binds nucleosome-free regions, but is not excluded from heterochromatin or DNase-inaccessible DNA

Unlike bacteria, yeast genomes are packaged into nucleosomes. Thus, we wanted to assess whether GapR-seq is impacted by the presence of nucleosomes and, more generally, whether GapR can report on positive supercoiling in both eu- and hetero-chromatin. We first examined GapR binding in heterochromatin, such as the yeast mating cassettes, and found that GapR can still access these relatively compacted loci (*Figure 5—figure supplement 1D*). To directly interrogate how nucleosomes impact GapR binding, we computationally compared GapR-seq to nucleosome occupancy inferred from micrococcal nuclease footprinting (MNase-seq), in which nucleosome centers are marked by peaks in read coverage (*Cutler et al., 2018*). We found that nucleosomes are often in close proximity to GapR peaks (*Figure 5C*, *Figure 5—figure supplement 1D, E*), with positions of high GapR enrichment found within 200 bp of nucleosomes (*Figure 5D*, *Figure 5—figure supplement 1F*). We conclude that GapR can bind near nucleosomes and is not generally excluded from heterochromatic DNA.

We also compared GapR enrichment to DNase I hypersensitivity (DNase-seq) data, which probes general DNA accessibility (*Zhong et al., 2016*). Although there was some overlap between sites of GapR binding and DNase cleavage, there were many DNase-sensitive regions not bound by GapR, and many loci with high GapR ChIP that were not DNase-accessible (*Figure 5C*, *Figure 5—figure supplement 1D, E*), indicating that DNA accessibility is not predictive of GapR enrichment. We then generated heatmaps of DNase accessibility at genomic regions with the highest GapR enrichment (*Figure 5E*), and vice versa (*Figure 5—figure supplement 1G*, see Materials and methods). These

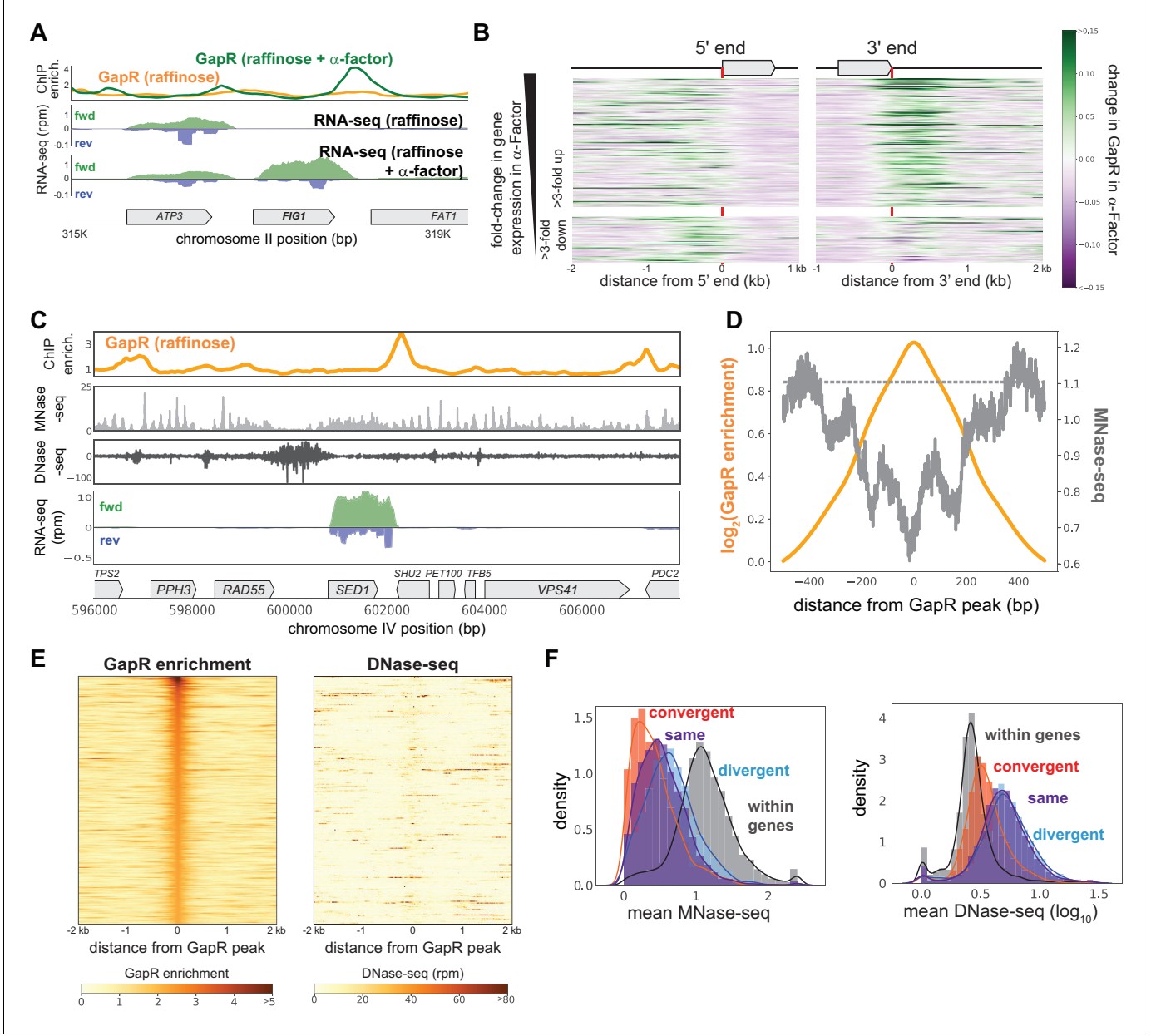

**Figure 5.** GapR binding in *S. cerevisiae* is responsive to transcription and is not restricted to open chromatin. (**A**) GapR enrichment at *Figure 1* in raffinose without (orange) or with (green) α-factor arrest before GapR induction (top). Transcription of the forward (green) and reverse (blue) strands in raffinose without (second panel) or with (third panel) α-factor arrest with annotated genes indicated. (**B**) Heatmap showing α-factor dependent change in GapR enrichment at the 5' and 3' ends of long transcription units (TUs) sorted by transcriptional change in α-factor. (**C**) GapR enrichment (orange) compared to nucleosome occupancy (MNase-seq, light gray) and chromatin accessibility (DNase-seq, dark gray). Transcription of the forward (green) and reverse (blue) strands with annotated genes indicated. (**D**) GapR chromatin immunoprecipitation (ChIP) peaks are de-enriched for nucleosomes. Log$_2$(GapR enrichment) (orange, left y-axis), MNase-seq reads (dark gray, right y-axis), and mean genomic MNase-seq occupancy (dashed gray line). (**E**) Heatmap of GapR enrichment (left) and DNase accessibility (right) of the 500 most GapR-enriched loci. (**F**) Association between transcriptional orientation and MNase-seq and DNase-seq.

The online version of this article includes the following figure supplement(s) for figure 5:

**Figure supplement 1.** GapR chromatin immunoprecipitation (ChIP) compared to nucleosome occupancy and chromatin accessibility.

heatmaps revealed that GapR peaks are not highly accessible to DNase and DNase-sensitive loci are not highly enriched for GapR. GapR-enriched and DNase-sensitive sites are nearly identical in AT content (*Figure 5—figure supplement 1H*), indicating that GapR is not excluded from DNase-sensitive regions due to base composition. Next, we examined DNase sensitivity and nucleosome binding in different transcriptional orientations. Although convergently transcribed regions have increased GapR occupancy (*Figure 4E*), these loci are less DNase-accessible and more nucleosome-free than divergently transcribed regions (Kolmogorov–Smirnov test, DNase-seq $p<10^{-131}$, MNase-seq $p<10^{-71}$; *Figure 5F*). Taken together, these results demonstrate that GapR prefers to bind in nucleosome-free regions, but DNA supercoiling, rather than chromatin accessibility, is primarily responsible for GapR occupancy.

## Comparison of GapR-seq and a psoralen-based method

Positive supercoiling has been previously examined in *S. cerevisiae* using a psoralen-based method in which positively supercoiled regions are inferred based on their reduced intercalation of psoralen (*Achar et al., 2020*). We directly compared our GapR-seq to this prior data, but found little overlap or correlation between GapR-enriched sites and those regions de-enriched for psoralen intercalation (*Figure 5—figure supplement 1I, J*). In contrast to that psoralen-based study, which suggested that positive supercoils accumulate within gene bodies and is not strongly dependent on transcription (*Achar et al., 2020*), our GapR-seq demonstrates a clear transcription-dependent 3' end bias.

## Positive supercoiling in yeast is associated with centromeres, pericentromeres, and cohesin

Collectively, our results show that GapR-seq maps where positive supercoils accumulate, such as the 3' ends of genes. We also asked if our GapR data captured positive supercoiling in other contexts. In yeast, positive supercoiling has been proposed to accumulate at centromeres, with supercoiling constrained within the centromeric sequences (CEN) and stabilized by binding of the CBF3 complex and the centromeric histone H3 variant, CENP-A/Cse4 (*Díaz-Ingelmo et al., 2015*; *Steiner and Henikoff, 2015*; *Verdaasdonk and Bloom, 2011*). Centromeric-positive supercoiling was not found in psoralen arrays (*Figure 6—figure supplement 1A*). In contrast, GapR accumulates at CEN upon α-factor arrest and when grown in glycerol, which extends G1 phase (*Figure 6A, B*, *Figure 6—figure supplement 1A*). By aligning GapR enrichment over all 16 CEN, we found that GapR occupancy was highest immediately to the 5' of CEN, upstream of the first centromere determining element and remained high ~500 bp to the 5' and 3' of CEN, with a small 3' shoulder (*Figure 6C*, *Figure 6—figure supplement 1B*). These data validate the notion, based on prior plasmid-supercoiling and in vitro studies, that positively supercoiled DNA is found within centromeres (*Díaz-Ingelmo et al., 2015*; *Steiner and Henikoff, 2015*).

Yeast pericentromeres are 10–30 kb cohesin-associated regions that flank centromeres (*Lawrimore and Bloom, 2019*). Cohesin is a chromosome organizing protein complex that mediates sister chromatid cohesion, homologous recombination, and other diverse functions by topologically linking distant loci (*Moronta-Gines et al., 2019*). Cohesin accumulates between convergent genes, including those that define pericentromere boundaries, and rapidly compacts positively supercoiled DNA in vitro, suggesting that cohesin may preferentially associate with such DNA (*Glynn et al., 2004*; *Lengronne et al., 2004*; *Paldi et al., 2020*; *Sun et al., 2013*). We investigated the relationship between positive supercoiling and cohesin localization by comparing our GapR data to previously published Scc1 (the kleisin subunit of cohesin) ChIP from cells arrested in metaphase (*Paldi et al., 2020*). In all media conditions, GapR was modestly to highly enriched between the convergent genes that mark pericentromere boundaries (*Figure 6D*, *Figure 6—figure supplement 1C, D*). Outside of pericentromeres, cohesin is also frequently, but not exclusively, associated with convergent genes. These cohesin-enriched regions were also bound by GapR (*Figure 6—figure supplement 1E*), supporting the idea that cohesin binding is associated with positive supercoiling.

To systematically examine any relationship between positive supercoiling and cohesin, we generated heatmaps of GapR enrichment surrounding the 500 highest cohesin-bound regions, finding that >90% of all cohesin peaks in glycerol had significant neighboring GapR enrichment within 200 bp (GapR enrichment $> \mu + \sigma$; *Figure 6E*). Conversely, when we examined cohesin enrichment surrounding the 500 highest GapR ChIP peaks, we found that positively supercoiled DNA was

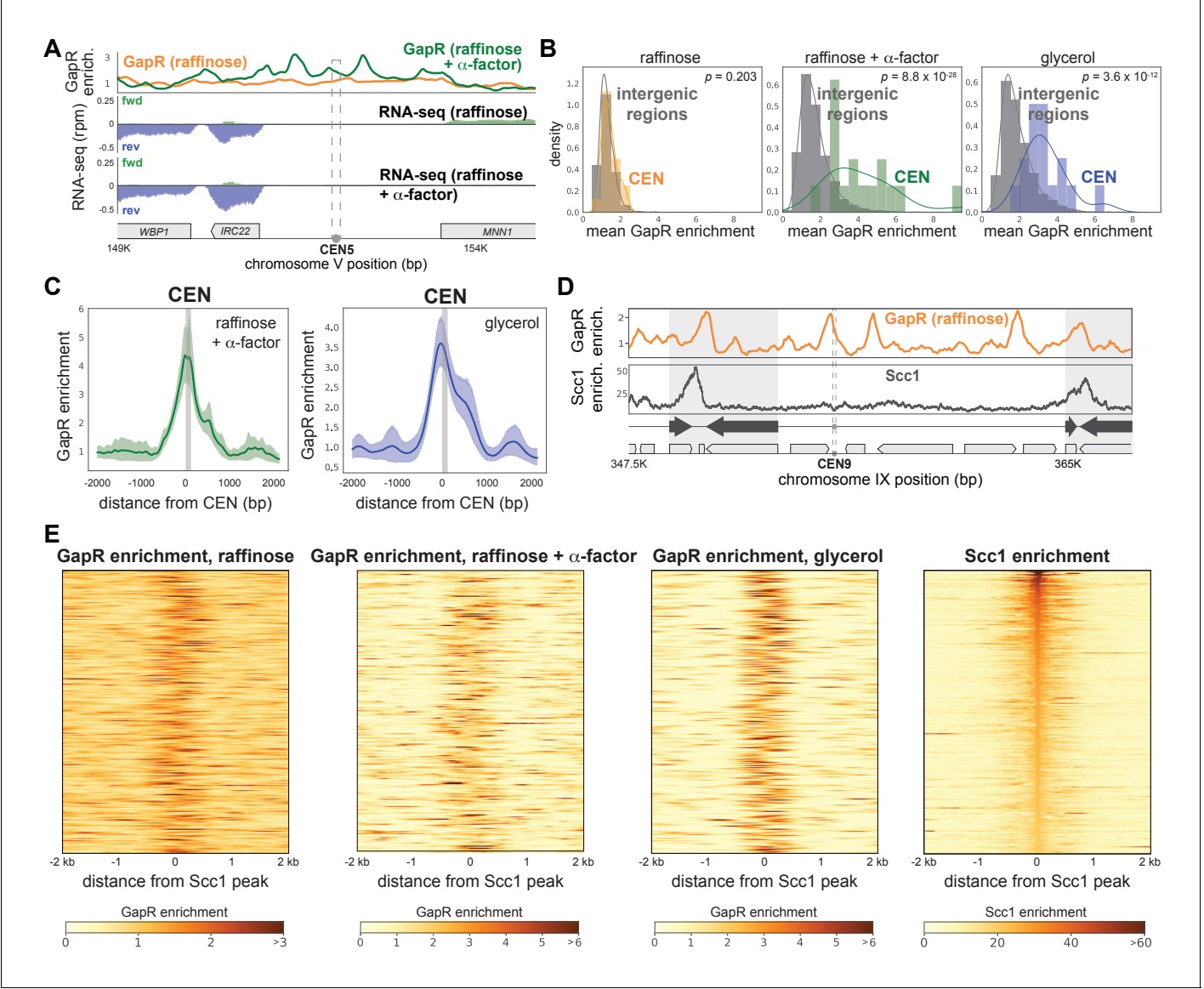

**Figure 6.** Positively supercoiled DNA is associated with centromeres and cohesin. (**A**) GapR enrichment at CEN5 in cells without (orange) or with (green) α-factor arrest (top). Transcription of the forward (green) and reverse (blue) strands in cells grown in raffinose without (second panel) or with (third panel) α-factor arrest with annotated genes indicated. (**B**) GapR enrichment at centromeres in cells grown in raffinose, after α-factor arrest, and grown in glycerol. Student's t-test p-value is shown. (**C**) GapR enrichment over all centromeres after α-factor arrest (green) or grown in glycerol (blue). Mean enrichment (solid line) with 95% confidence intervals (shaded area). Gray bar represents position of centromeres. (**D**) GapR (raffinose, top) and cohesin (Scc1 enrichment, bottom) are associated with convergent genes (arrows) at pericentromere boundaries (shaded areas). (**E**) Heatmaps of GapR (three left panels) and Scc1 (right) enrichment at the 500 most Scc1-bound loci.

The online version of this article includes the following figure supplement(s) for figure 6:

**Figure supplement 1.** GapR chromatin immunoprecipitation (ChIP) at centromeres, pericentromeres, and cohesin-bound regions.

frequently, but not always, associated with strong cohesin binding (*Figure 6—figure supplement 1F*). Our results support the notion that positive supercoiling influences cohesin localization. More broadly, our findings (1) validate the idea that positive supercoils are a key feature of centromeres, pericentromeres, and cohesin-binding sites, and (2) that GapR-seq reveals, with high resolution, the positions of these supercoils within the yeast genome.

## Positive supercoiling is found within the rDNA locus and at autonomously replicating sequences

In addition to centromeres and cohesin-binding sites, we also observed GapR enrichment within the 150–200 tandem repeats of ribosomal DNA (rDNA) found on the right arm of chromosome XII (*Figure 7A*). GapR binds at two major peaks in the rDNA locus: (1) one in the unique region at the 3' end of the rDNA locus, which likely arises from transcription of the last 35S rDNA repeat, and (2) one within the rDNA, ~1600 bp upstream of the ribosomal autonomously replicating sequence (rARS) that coincides with the replication fork barrier (RFB), with an additional minor peak over the rARS (*Figure 7A*). These two peaks manifest in all media conditions, but are most prominent in α-factor-arrested and glycerol-grown cells (*Figure 7A*, *Figure 7—figure supplement 1A*), and do not result from changes in rDNA copy number (*Figure 7—figure supplement 1A*). The RFB is an ~100 bp sequence at the 3' end of the 35S rDNA where Fob1p binds to block replisome progression and

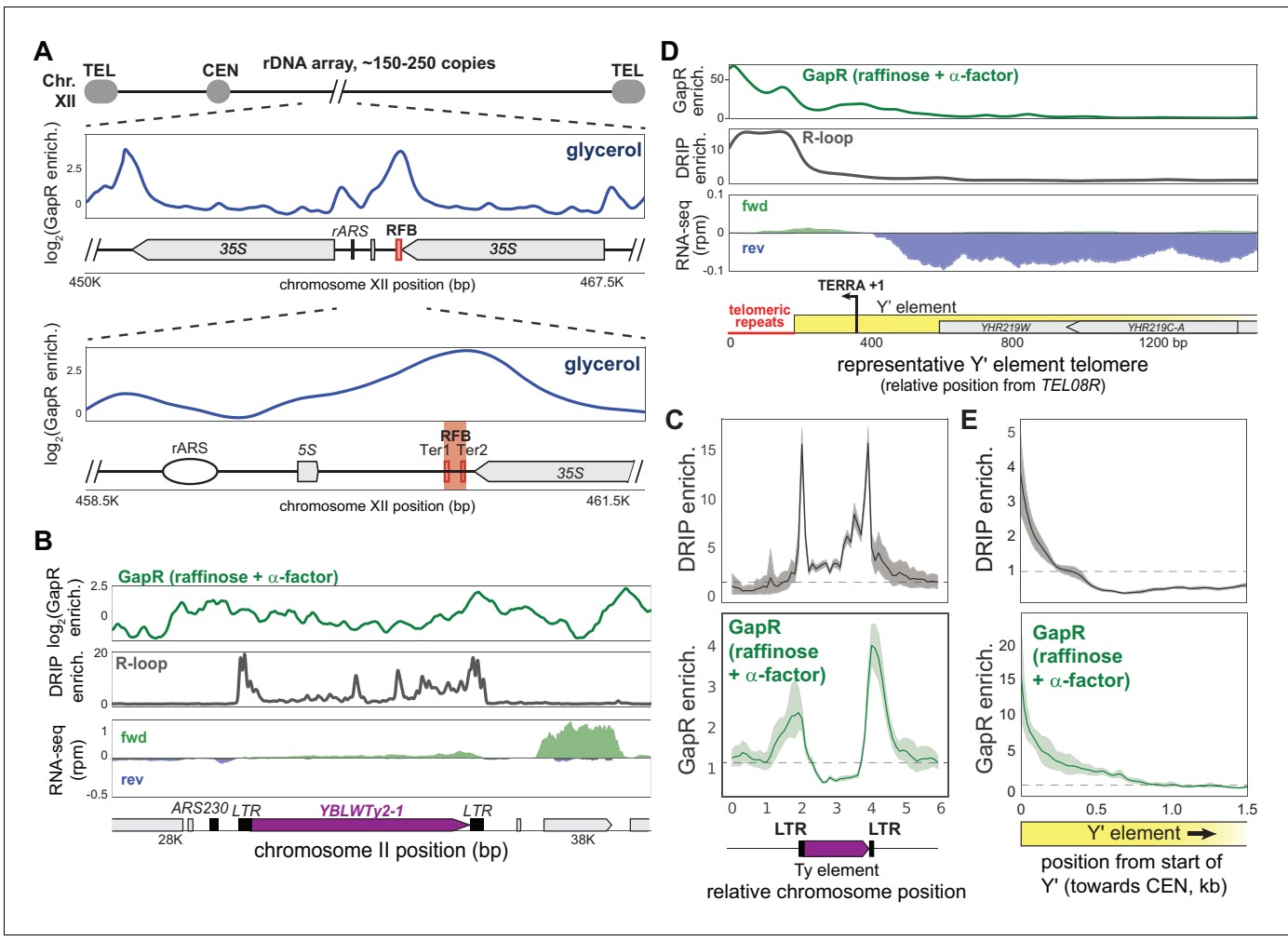

**Figure 7.** Positive supercoiling is associated with autonomously replicating sequences (ARS) and R-loops. (A) GapR enrichment (glycerol) at the rDNA shown in two successive zoom-ins (top and bottom panels) with the replication fork barrier (RFB) and termination sequences (Ter1/Ter2) indicated. (B) Log₂(GapR enrichment) (α-factor) at a Ty element (top). DNA-RNA hybrid formation by DRIP-seq (middle). Transcription of the forward (green) and reverse (blue) strands with annotated genes indicated (bottom). (C) Alignment of GapR and DRIP enrichment surrounding all yeast Ty elements. Data indicate mean (solid line) with 95% confidence intervals (shaded area), no enrichment (dotted line). (D) GapR enrichment at a telomere with a Y' element (top). DRIP enrichment (middle). Transcription of the forward (green) and reverse (blue) strands with the organization of the Y' element indicated (bottom). Position given is from end of *TEL08R*. (E) Alignment of GapR and DRIP enrichment surrounding all yeast telomeres with Y' elements as in (C). Telomeric repeats are removed from analysis.

The online version of this article includes the following figure supplement(s) for figure 7:

**Figure supplement 1.** GapR chromatin immunoprecipitation (ChIP) at rDNA, autonomously replicating sequences, and R-loops.

prevent collisions between the replication fork and RNA polymerase transcribing the 35S rDNA (*Brewer and Fangman, 1988*; *Kobayashi and Horiuchi, 1996*; *Kobayashi, 2003*). The GapR-seq signal was centered over the Fob1p binding sites (Ter1 and Ter2) within the RFB, precisely where the replication machinery would arrest, and to the 5′ of rARS (*Figure 7A*, *Figure 7—figure supplement 1A*). Because of the GapR enrichment near rARS, we asked whether GapR was enriched near other ARS and found that GapR was enriched within many ARS compared to intergenic sequences in cells treated with α-factor and grown in glycerol (*Figure 7—figure supplement 1B, C*). Given that α-factor arrest and glycerol growth both lead to an extended G1 phase, DNA replication is likely not responsible for the positive supercoiling at these regions. Instead, this accumulation could be due to transcriptional effects or proteins bound to ARS (e.g., pre-replicative complex) and RFB that act as barriers to supercoiling diffusion.

## Positive supercoiling is associated with R-loops

Our α-factor and glycerol GapR-seq datasets also revealed many strong peaks associated with retro-transposable (Ty) elements, usually with highest enrichment over the terminal repeats (LTR) that flank Ty elements (*Figure 7B*, *Figure 7—figure supplement 1D*). These peaks were especially striking in the vicinity of poorly expressed Ty elements or divergently oriented regions next to Ty elements (*Figure 7B*, *Figure 7—figure supplement 1D*) because transcription-dependent positive supercoiling does not occur in this context. The LTRs of Ty elements are associated with stable DNA-RNA hybrids (R-loops) and have been mapped by DRIP-seq, which uses the S9.6 antibody to specifically recognize DNA-RNA hybrids (*Chan et al., 2014*; *El Hage et al., 2014*; *Niehrs and Luke, 2020*; *Wahba et al., 2016*). To compare our GapR-seq to published S1-DRIP-seq data (*Wahba et al., 2016*), we aligned all 49 yeast Ty elements and examined R-loop formation and GapR binding. We observed two peaks of DNA-RNA hybrids centered on the LTRs of Ty elements, with the peaks of GapR centered just beyond each DRIP-seq peak (*Figure 7C*, *Figure 7—figure supplement 1E*). These results suggest that positive supercoils are associated with R-loops.

DNA-RNA hybrids also occur at telomeres, where transcription of telomeric sequences produces a long, noncoding telomeric repeat-containing RNA (TERRA) that invades telomere DNA and mediates telomere maintenance (*Balk et al., 2013*; *Graf et al., 2017*; *Luke and Lingner, 2009*; *Niehrs and Luke, 2020*). Yeast telomeres are highly repetitive so many telomere ends are incompletely sequenced, but each typically consists of a telomeric repeat and an X element, with ~50% of telomeres also containing one or more Y′ elements (*Louis, 1995*). To assess GapR and R-loop enrichment in these regions, we assigned reads mapping to repeat sequences randomly across copies of that repeat, allowing for analysis of these repetitive sequences in aggregate. For telomeres containing Y′ elements, we observed DNA-RNA hybrids coincident with the telomeric repeats and where TERRA transcription occurs (*Figure 7D*; *Pfeiffer and Lingner, 2012*). Notably, GapR is also highly enriched at these telomeres, with enrichment greatest over the telomeric repeats and remaining high, ~500 bp towards centromeres, past the DNA-RNA hybrids (*Figure 7D*, *Figure 7—figure supplement 1F, G*). Although some transcription does occur near and within Y′ elements, we find that GapR enrichment is much higher in magnitude at telomeres than other transcribed regions (compare *Figure 7D* with *Figure 5A*), suggesting that transcription cannot fully explain GapR binding at these loci. We then examined telomeres containing X elements but not Y′ elements and found that GapR and R-loops are enriched at these telomeres as well (*Figure 7—figure supplement 1F, G*). Because R-loops occur when TERRA invades and unwinds a DNA duplex, these R-loops likely produce hypernegatively supercoiled regions of DNA and may be accompanied by the compensatory structuring of overtwisted, positively supercoiled DNA that balances torsional stress (*Figure 7—figure supplement 1H*). Local overtwisted DNA could also act as a barrier to prevent further expansion of R-loops (*Figure 7—figure supplement 1H*). Taken all together, our results indicate that positive supercoils are features of many chromosomal loci in yeast. More broadly, we propose that GapR-seq is a flexible and powerful new approach for probing positive supercoiling in cells, from bacteria to eukaryotes.

## Discussion

The pervasiveness, chromosomal context, and consequences of supercoiling remain poorly understood, in part because methods to map positive supercoiling in vivo at high resolution have been

lacking. Here, we developed a method to interrogate positive supercoiling in both bacterial and eukaryotic cells using GapR, a protein sensor of overtwisted DNA. Using single-molecule MT experiments, we demonstrated that GapR preferentially and stably binds overtwisted DNA. Consequently, GapR localizes to overtwisted, positively supercoiled DNA in bacteria and yeast, allowing positive supercoils to be systematically identified by GapR chromatin immunoprecipitation sequencing (GapR-seq). This new method revealed that positive supercoiling is a pervasive feature of genomes, with remarkably similar patterns documented in bacteria and yeast. Positive supercoils accumulate in a transcription-dependent manner at the 3' ends of genes and are particularly enriched in regions where transcription is convergent. In yeast, GapR-seq further revealed that positive supercoils are associated with centromeres, cohesin-binding sites, ARS, and DNA-RNA hybrids (R-loops), suggesting that positive supercoils may have regulatory or structural roles in each of these chromosomal elements.

## GapR is a sensor for overtwisted DNA

Other proteins are known to interact preferentially with positively or negatively supercoiled DNA (*Ding et al., 2014*; *Sun et al., 2013*; *Zorman et al., 2012*), but, to our knowledge, GapR-DNA interactions are unique in that they are destabilized by negative supercoiling, leading to a significant preference for overtwisted DNA (*Figure 1E–H*). Although GapR interacts with a range of DNA structures in vitro (*Huang et al., 2020*; *Tarry et al., 2019*), our MT data and topological assays (*Guo et al., 2018*) indicated that GapR interacts most stably with overtwisted conformations of DNA (*Figure 1A, D*). We propose that GapR engages in cycles of sliding, hopping, and partial dissociation, along with reorganization of plectonemic supercoiling and partially strand-separated regions when in complex with relaxed or undertwisted DNA, explaining the dynamics observed in our MT experiments (*Figure 1E*). We note that for negative supercoiling DNA readily strand-separates when under moderate tension (>0.5 pN) (*Meng et al., 2014*; *Strick et al., 1998*), with 'phase coexistence' of plectonemic, extended, and strand-separated DNA occurring for tensions slightly above those occurring in physiological DNA supercoils (*Marko, 2007*). It is likely that GapR, by forcing DNA to overtwist, shifts the region of phase coexistence of B-form, plectonemic supercoiled, and strand-separated DNA down to the 0.3 pN used here, leading to large extension fluctuations (*Figure 1E*, *Figure 1—figure supplement 1G*). As has been observed for other DNA nucleoid-structuring proteins (*Kamar et al., 2017*; *Skoko et al., 2004*), complete dissociation of GapR from DNA to solution is remarkably slow, indicating that the dynamics of extension fluctuations are dependent on DNA dynamics and local GapR-DNA dynamics (hopping or sliding) rather than on complete GapR dissociation from DNA (*Figure 1—figure supplement 1H, I*). For this study, the key results are that the peaks of the 'hat' curves are at positive σ, and that GapR-DNA structures are dynamically far more stable for positive σ than for negative σ, indicating greater stability of GapR binding to positive supercoils relative to negative supercoils.

We exploited the binding preference of GapR to develop it as a generic sensor of overtwisted DNA. There are several caveats to our GapR-seq approach. (1) GapR recognizes overtwisted DNA and, in our topological assays, binds to ~8.5° of twist (*Guo et al., 2018*). GapR may not be able to recognize regions more modestly twisted or more highly twisted, or if writhed structures form (e.g., plectonemic or solenoidal supercoils) that are somehow constrained from interconverting to a twisted form. (2) AT-rich DNA can adopt intrinsically bent structures with narrowed minor grooves that may be recognized in a supercoiling-independent manner by GapR (*Arias-Cartin et al., 2017*; *Guo et al., 2018*; *Haran and Mohanty, 2009*; *Huang et al., 2020*; *Ricci et al., 2016*). In GC-rich organisms such as *Caulobacter*, the association with AT-rich DNA was more pronounced than in *E. coli* or yeast (*Figure 2—figure supplement 1F*, *Figure 4—figure supplement 1E*). (3) The dynamics of GapR-DNA exchange in vivo are unknown, as is whether GapR binding affects the kinetics or activity of heterologous topoisomerases. Additional studies are needed to fully understand the implications of GapR binding and the in vivo structures recognized by GapR.

For the case of eukaryotic chromatin, one might be concerned about the effects on GapR-seq of the low stiffness to positive torsional stress observed in single-molecule studies (*Le et al., 2019*), but our observations of similar GapR-seq patterns in bacteria and in yeast suggest that GapR-seq is working similarly in yeast chromatin as in bacterial chromosomes. The likely reason for this is that because GapR binds to overtwisted DNA, GapR binding is insensitive to what the large-scale conformation of the DNA region is, be it plectonemic supercoils (as occurs in bacteria) or locally deformed

nucleosome structures (thought to be the case in chromatin; *Le et al., 2019*), and is instead sensing the stress (DNA torque). Positive torque in the DNA is a source of energy that drives GapR binding, regardless of the large-scale conformational response of the chromosomal region under that positive torsional stress. This in turn suggests that regions of positive torsional stress in yeast chromatin likely have similar levels of DNA torque to those found in bacteria, which is logical since the physical processes generating those torques are similar.

## Positive supercoiling is pervasive and recapitulates the 'twin-domain' model

Transcription leads to the formation of positive and negative supercoils ahead of and behind, respectively, the transcription bubble, referred to as the 'twin-domain' model (*Figure 2A*; *Liu and Wang, 1987*). The existence of transcription-dependent supercoiling in vivo has been confirmed indirectly in numerous ways (*Nelson, 1999*), including measuring transcription-dependent changes in plasmid linking number (*Drlica et al., 1988*; *Wu et al., 1988*) and interrogating the effects of topoisomerase inhibition (*Khodursky et al., 2000*). More recently, psoralen, which preferentially binds and crosslinks to negatively supercoiled DNA, has been used to probe supercoiling genome-wide. These studies indicated that negative supercoiling is pervasive, transcription-dependent, and enriched at promoters, consistent with the twin-domain model (*Achar et al., 2020*; *Kouzine et al., 2013*; *Naughton et al., 2013*; *Teves and Henikoff, 2014*). However, one study suggested that negative supercoils also arise downstream of transcribed genes with positive supercoils accumulating in intragenic regions regardless of transcriptional activity (*Achar et al., 2020*), findings at odds with the twin-domain model. In contrast, GapR-seq revealed, in both bacteria and yeast, that positive supercoiling is (1) in intergenic or transcriptionally silent regions that lie at the 3′ ends of transcribed genes, and not generally within gene bodies, (2) depleted at the 5′ ends of genes, (3) transcription-dependent, with signal roughly proportional to the transcriptional activity of upstream genes, and (4) trapped by convergent transcription, all as predicted by the twin-domain model. Unlike psoralen approaches that infer positive supercoiling based on the absence of psoralen crosslinking, GapR-seq specifically and directly probes for positively supercoiled DNA.

The ability to detect positive supercoils using GapR-seq in both bacteria and yeast indicates that positive supercoils are not fully, or at least immediately, dissipated by topoisomerases in vivo. GapR-seq also allows mapping of positive supercoiling at high (<1 kb) resolution, demonstrating that positive torsion appears capable of diffusing over a few kb (*Figure 2—figure supplement 1L*, *Figure 4—figure supplement 1G*). However, supercoil diffusion is limited by transcription as GapR signal was rare within transcribed regions (*Figures 3F* and *4E*). Supercoiling may also be limited by the binding of DNA structuring proteins like nucleosomes or other complexes (*Figures 5D* and *7A*). Finally, because the distribution of positive supercoils downstream of actively transcribed genes was consistent between bacteria and yeast, we anticipate that similar patterns are likely to be found in other organisms as well.

## Positive supercoiling in chromosome organization

GapR-seq suggested an association between positive supercoiling and yeast centromeres, cohesin-binding sites, ARS, and R-loops, revealing potentially significant roles for positive supercoils in genome organization and function. Interestingly, these associations were strongest in conditions where yeast were primarily in G1 phase, suggesting that active replication may clear GapR from the DNA or that rapid growth diminishes the deposition of GapR on chromosomes, dampening signal. For centromeres, prior work suggested that the intrinsic architecture and assembly of the CENP-A histone complex at centromeres leads to positive supercoiling (*Díaz-Ingelmo et al., 2015*). Additionally, positive supercoiling has been proposed to aid in cohesin deposition (*Sun et al., 2013*). Our results support these ideas and now provide insight into the precise localization of positive supercoils at these chromosomal regions (*Figure 6*). In higher eukaryotes, cohesin is also found outside of centromeres where it accumulates at some CTCF sites to extrude loops and form topologically associated domains (TADs) (*Fudenberg et al., 2017*; *Nora et al., 2017*; *Rao et al., 2017*). Given the association between cohesin and positive supercoils documented here, we suggest that GapR-seq may be particularly useful in probing the contribution of positive supercoiling to TAD formation.

Positive supercoiling associated with ARS and R-loops has not been well characterized or carefully probed, again because of limited methods for mapping supercoils in vivo. We propose that positive supercoiling near the rARS and other ARS may result from enhanced trapping of topological stress in these regions. Positive supercoiling occurs in replication-transcription encounters (*García-Muse and Aguilera, 2016*), and our data raise the possibility that supercoiled DNA could be trapped between the converging replication and transcription machineries in yeast. Our results also indicate that positive supercoils occur adjacent to, but do not fully overlap with, R-loops such as the boundaries of Ty elements and telomeres (*Figure 7B–E*). The noncoding TERRA invades telomeric DNA to form R-loops and promote telomere maintenance in eukaryotes (*Balk et al., 2013*; *Bettin et al., 2019*). Because every ~10 bp captured in an R-loop represents one negative supercoil, DNA-RNA hybrids like TERRA are potential reservoirs of extreme negative superhelicity. Recent work has demonstrated that R-loops may be extremely sensitive to supercoiling as opening an R-loop in relaxed, topologically constrained DNA leads to the formation of positive supercoiling elsewhere, which can impede R-loop formation (*Stolz et al., 2019*). We propose that positive supercoiling may be generated during R-loop formation (*Figure 7—figure supplement 1H*) and that positive supercoiling adjacent to TERRA hybrids could be a barrier to further melting of the DNA duplex and R-loop spreading (*Figure 7—figure supplement 1H*). Recent studies have suggested that an overabundance of telomeric R-loops causes replicative stress and increased recombination rates in human cells, with these general pathways conserved in yeast (*Pan et al., 2019*; *Petti et al., 2019*). Further work will be needed to dissect how positive supercoils arise near R-loops and how they impact genome structure and function.

In sum, our GapR-seq approach provides high-resolution, genome-wide maps of positive supercoils in both bacteria and yeast. These maps reveal the extent and distribution of both transcription-induced positive supercoils as well as supercoiling in other genomic contexts, such as centromeres and telomeres, where positive supercoils may play important roles in genome organization. We anticipate that our GapR-seq method will be easily extended to diverse bacterial and eukaryotic organisms for probing the origins and consequences of DNA torsion and understanding how DNA topology impacts gene expression, chromosome structure, and genome maintenance.

## Materials and methods

### Growth conditions and chemical treatments

*E. coli* strains were grown in LB (10 g/L NaCl, 10 g/L tryptone, 5 g/L yeast extract) at 37°C with shaking at 200 rpm unless noted. Antibiotics were supplemented as needed (carbenicillin: 50 µg/mL liquid/100 µg/mL plate, and kanamycin: 30 µg/mL/50 µg/mL). The $P_{tet}$ promoter was induced by supplementing with 25 ng/mL anhydrous tetracycline (aTc, diluted in water) for 2 hr. For transcriptional inhibition experiments, rifampicin 300 µg/mL was added for 20 min before fixation and ChIP. *S. cerevisiae* strains were grown in YPD, YEP + 2% glycerol, YEP + 2% raffinose, or in SC-LEU as appropriate. The $P_{gal1-10}$ promoter was induced by addition of 2% galactose for 6 hr. For G1 arrest, α-factor was added to 1 µg/mL in YEP + 2% raffinose for 2 hr before addition of galactose for 6 hr before cell harvest. Optical density was measured at 600 nm using a Genesys 10 Bio Spectrophotometer or in a Synergy H1 plate reader.

### Strain construction

*E. coli* strains were derivatives of MG1655 K-12 (ML6) or BL21(DE3). Strain ML3323 was constructed by electroporation of pKS22b-hSUMO-GapR[1-76] into BL21(DE3). Strain ML3324 was constructed by electroporation of pKS22b-hSUMO-GapR[1-81] into BL21(DE3). Strain ML3284 was constructed by electroporation of pGapR-3xFLAG into ML6. Strain ML3285 was constructed by electroporation of pGapR[WT] into ML6. Strain ML3286 was constructed by electroporation of pGapR-3xFLAG into ML6. Strain ML3325 was constructed by electroporation of pGapR[1-76]-3xFLAG into ML6.

*S. cerevisiae* strains were derivatives of OAY470 (ML3287, gift from S. Bell lab). Strain ML3288 was constructed by transforming PmeI cut pNH605-gal1/10-GapR[WT] into ML3287 and selecting on SC-LEU plates. Strain ML3289 was constructed by transforming PmeI cut pNH605-gal1/10-GapR-3xFLAG into ML3287 and selecting on SC -LEU plates.

## Plasmid construction

### *E. coli* plasmids

Expression plasmids pKS22b-hSUMO-GapR$^{1-76}$were constructed by amplifying *C. crescentus* genomic DNA with primers pKS22-GapR_NcoI and pKS22-GapR1-76_NotI, and pKS22b-hSUMO-GapR$^{1-81}$ was constructed by amplifying *C. crescentus* genomic DNA with primers pKS22-GapR_NcoI and pKS22-GapR1-81_NotI. These PCR products were digested with the NcoI and NotI and ligated into pKS22b digested with the same enzymes. pGapR$^{WT}$ was constructed by amplifying *C. crescentus* genomic DNA with primers pKVS45-gapR-f and pKVS45-gapR-r, and pGapR-3xFLAG was constructed by amplifying *C. crescentus* genomic DNA hosting GapR-3xFLAG with primers pKVS45-gapR-f and GapR-pKVS_3xF_r. These PCR products were digested with the restriction enzyme BsmBI and ligated into pKVS45 digested with the same enzyme. pGapR$^{1-76}$-3xFLAG was constructed from pGapR-3xFLAG by round the horn mutagenesis using phosphorylated primers GapR1-76_l and GapR3xF_r.

### Yeast integration plasmids

pNH605-gal1/10-GapR$^{WT}$ was constructed by performing splice-overlap-extension (SOE) PCR with a fragment containing the Gal1-10 promoter generated by PCR with primers Gal1/10_ClaI_f and Gal1/10_ClaI_r and a fragment containing GapR$^{WT}$ generated with primers GapR_fwd and GapR_Xho_r3. The resulting SOE product was digested with ClaI and XhoI and ligated into pNH605 digested with the same restriction enzymes. pNH605-gal1/10-GapR-3xFLAG was constructed by performing SOE PCR with a fragment containing the Gal1-10 promoter generated by PCR with primers Gal1/10_ClaI_f and Gal1/10_ClaI_r and a fragment containing GapR-3xFLAG generated with primers GapR_fwd and GapR_Xho_r. The resulting SOE product was digested with ClaI and XhoI and ligated into pNH605 digested with the same restriction enzymes.

## Purification of GapR

GapR was purified as reported previously (*Guo et al., 2018*) with the following modifications: 1 L of His$_7$-SUMO-GapR expressing BL21(DE3) cells were grown in 2xYT to OD$_{600}$ ~ 0.4–0.5 at 37°C and induced with 0.4 mM IPTG for 18–20 hr at 18°C. Cells were then harvested by centrifugation and resuspended in 40 mL buffer A (50 mM sodium phosphate [pH 7.5], 0.5 M NaCl, 20 mM imidazole) supplemented with a SIGMA*FAST* Protease Inhibitor Tablet (Sigma) and benzonase (Sigma). The cell resuspension was then lysed using a Microfluidizer (15,000 psi, five passes). The cell debris was cleared by centrifugation for 30 min at 10,000 × *g* and passed over Ni-NTA agarose resin (QIAGEN) pre-equilibrated with buffer A at 4°C. Resin was washed with buffer A, then with A containing 100 mM imidazole. GapR was then eluted with buffer A containing 500 mM imidazole. SUMO protease was then added and SUMO cleavage proceeded overnight at 4°C with dialysis into fresh buffer A. To remove uncut His$_7$-SUMO-GapR and SUMO protease, protein was then passed over Ni-NTA agarose resin, collecting the flowthrough, and then an additional column volume of buffer A was passed and collected from the column. The flowthrough was then diluted twofold and then directly applied to a HiTrap Heparin HP (5 mL) (GE Healthcare) column, pre-equilibrated in buffer B (50 mM sodium phosphate [pH 7.5], 10 mM NaCl). GapR was eluted with a two-step elution (buffer B + 0.5 M NaCl and buffer B + 1.0 M NaCl), each step being five column volumes. 1 M NaCl fractions containing GapR were collected and concentrated on an Amicon 3K Centrifugal Filter Unit (Millipore) and buffer exchanged into storage buffer (50 mM sodium phosphate [pH 7.5], 200 mM NaCl, 10% [v/v] glycerol) by gel filtration using a Superdex 200 Increase 10/300 GL column (GE Healthcare). Fractions containing GapR were identified by SDS-PAGE/Coomassie staining, concentrated, snap-frozen, and stored at −80°C.

GapR$^{1-76}$ was purified as with GapR with the following modifications: 0.2 L of His$_7$-SUMO-GapR$^{1-76}$ expressing BL21(DE3) cells were grown in 2xYT to OD$_{600}$ ~ 0.4–0.5 at 37°C and induced with 0.4 mM IPTG for 18–20 hr at 18°C. Cells were then harvested by centrifugation and resuspended in 40 mL buffer A (50 mM sodium phosphate [pH 7.5], 0.5 M NaCl, 20 mM imidazole) supplemented with a SIGMA*FAST* Protease Inhibitor Tablet. The cell resuspension was then lysed by sonication (fve cycles of 30 s on, 30 s off at 40 power on a Qsonica Q700). The cell debris was cleared by centrifugation for 10 min at 10,000 × *g* followed by centrifugation for 30 min at 30,000 × *g*, and passed over Ni-NTA agarose resin. Resin was washed with buffer A, then eluted stepwise with 40, 60, 80, 100,

and 500 mM imidazole. 80–500 mM imidazole fractions were combined and applied directly to a HiTrap Heparin HP (1 mL) (GE Healthcare) column, pre-equilibrated in buffer B (50 mM sodium phosphate [pH 7.5], 10 mM NaCl). His$_7$-SUMO-GapR truncations were washed with 5 mL 10 mM NaCl buffer B, then 2.5 mL 0.5 M NaCl buffer B, then eluted with 2.5 mL 1.0 M NaCl buffer B. 1 M NaCl fractions were dialyzed overnight in the presence of SUMO protease into fresh buffer A. To remove uncut His$_7$-SUMO-GapR$^{1-76}$, His$_7$-SUMO, and SUMO protease, dialyzed protein was passed over fresh Ni-NTA agarose resin and washed with an additional column volume of buffer A, collecting the flowthrough throughout. The flowthrough was then applied to a HiTrap Heparin HP (1 mL) column and processed as before. 1 M NaCl fractions containing GapR$^{1-76}$ were collected and concentrated on an Amicon 3K Centrifugal Filter Unit (Millipore) and exchanged into storage buffer (50 mM sodium phosphate [pH 7.5], 200 mM NaCl, 10% [v/v] glycerol) by gel filtration using a Superdex 200 Increase 10/300 GL column. Fractions containing GapR$^{1-76}$ were identified by SDS-PAGE/Coomassie staining, concentrated, snap-frozen, and stored at −80˚C.

## DNA topology assays

For DNA topology assays, nicked pUC19 was generated from negatively supercoiled pUC19 (NEB) by treatment with Nt.BspQI (NEB) followed by PCR purification. Mixtures of GapR and nicked pUC19 DNA (40 ng) in 1× T4 DNA ligase buffer were incubated at 30˚C for 60 min. When used, T4 DNA ligase (NEB) was diluted to 1 U/mL in 1× T4 DNA ligase buffer and 1 U was added to reactions and incubated for an additional 1.5 hr at room temperature (RT). Reactions were stopped by addition of 1% SDS and 10 mM EDTA (final concentration), and digested with Proteinase K (NEB) for 1 hr at 37˚C. DNA loading buffer was added and samples electrophoresed. For one-dimensional electrophoresis, 1% TBE agarose gels were run at 130 V for 90 min and then in SYBR Gold and imaged with a Typhoon FLA 9500 or Azure Sapphire imager. For two-dimensional chloroquine gels, electrophoresis was performed by first running reactions on a 1% TBE agarose gel at 130 V for 90 min, then soaking the gel for 2 hr with shaking in 1× TBE supplemented with 4.5 mg/mL chloroquine phosphate (Santa Cruz Biotech). The gel was then turned 90˚ and electrophoresed in the orthogonal direction at 130 V for 60 min in 1× TBE supplemented with 4.5 mg/mL chloroquine phosphate. Chloroquine is a DNA intercalator that introduces (+) supercoils. In chloroquine, (-) supercoiled plasmids will become more relaxed, and migrate more slowly, whereas (+) supercoiled DNA will be further compacted, increasing its migration speed. The gel was washed 4 × 20 min in distilled water to remove chloroquine, stained with SYBR Gold for 2 hr, and imaged with a Typhoon FLA 9500 or Azure Sapphire imager. Relaxed plasmid standard was generated with *E. coli* Topoisomerase I (NEB). Positively supercoiled standard was generated with *Archaeoglobus fulgidus* reverse gyrase (*Guo et al., 2018*). A magnesium-dependent nuclease activity was detected in the DNA topology assays (*Figure 1—figure supplement 1A*), which is likely due to the benzonase added during the purification step.

## MT assays

MT assays were performed following a previously described protocol (*Giuntoli et al., 2015*) with minor modifications. Briefly, experiments were carried out using the 9702 bp plasmid pNG1175, which was linearized with SpeI and ApaI restriction enzymes (*Bai et al., 2011*). The linear molecule was ligated to ~900 bp PCR products carrying either biotinylated or digoxigenin-labeled nucleotides, with SpeI- and ApaI-compatible ends, respectively, resulting in a 11.4 kb long DNA fragment (leading to ~10 kb DNA tethers once attached to the flow cell).

Experiments were performed in flow cells of approximately 30 µL volume that were constructed by sandwiching Parafilm (Sigma-Aldrich) between coverslips and glass slides. Flow cells were functionalized by overnight incubation at 4˚C using 100 µL of 0.1 mg/mL anti-digoxigenin antibody (Roche) in 1× PBS to which 1 µL of 3 µm polystyrene bead stock (Polysciences) had been added, with the cover slip side down. The polystyrene particles adsorb permanently to the cover slip, serving as reference beads to determine the position of the glass surface. Functionalized flow cells were then passivated by incubation with 1% BSA and 1% F127 (Sigma-Aldrich) in 1× PBS overnight at 4˚C to block non-specific binding. To bind biotinylated DNA to beads, 10 ng of end-labeled pNG1175 DNA was mixed with 1 µL streptavidin-functionalized paramagnetic beads (M-270 Dynabeads, diluted 1:6 in PBS with 0.1% BSA; Invitrogen) in 10 µL 1× PBS and incubated for 10 min at RT. The bead-bound DNA was then diluted with 100 µL 1× PBS and introduced into the flow cell, with the

flow cell inverted so that the beads fell to the cover glass surface. The flow cell was incubated for 10 min at RT to allow the digoxigenin-functionalized DNA ends to bind the anti-digoxigenin-functionalized coverslip.

The assembled flow cell was then placed on a magnetic tweezers microscope setup, consisting of a 100× 1.35 NA (Olympus) microscope objective on a piezoelectric positioner (Piezojena), with permanent neodymium magnets that are positioned using a stepper-motor-driven translator as previously described (*Figure 1—figure supplement 1A*; *Giuntoli et al., 2015*; *Skoko et al., 2004*). Movement of the piezoelectric positioner and the consequent force applied to the DNA is controlled by moving the magnets closer or further from the flow cell. The relative positions of the reference beads and DNA-tethered bead are measured using an algorithm that uses the degree of focus of the beads to determine their distance from the glass surface. Labview (National Instruments) is used to control the vertical (z) position and rotation of the magnets, track the x–y motions of the beads, measure the z position of the beads, and calculate the forces on the beads as previously described (*Skoko et al., 2004*).

At the start of each experiment, beads were tested to identify a supercoilable DNA molecule for further study (non-nicked single dsDNA that is tethered to the bead). First, the DNA molecule was rotated with the magnet to verify that the extension length of the DNA changed significantly upon rotation, indicating that the DNA is supercoilable. Next, the DNA length was measured under a variety of forces to verify that the given bead was attached by a single molecule of DNA. An initial test is that the extension length under high forces is ~2.8 µm, the expected length for 11.4 kb DNA. Next, extension and force measurements at five magnet positions were performed to verify that the apparent persistence length of the candidate molecule is as expected for a singly tethered DNA.

Experiments were performed in GapR assay buffer (40 mM sodium phosphate [pH 7.5], 100 mM NaCl). The 'rotation extension' experiments were performed as follows. First, naked DNA extensions were measured against torque by rotating the magnet to twist DNA between σ = −0.03 and +0.03 at 0.3, 0.5 and 1.0 pN of magnetic forces. Next, we repeated the same series of measurements under the presence of GapR. For the rotation extension hysteresis experiment, one GapR-bound DNA was repeatedly turned from 0.0 σ to +0.03 σ down to −0.03 σ and back up to +0.03 σ in 0.005 σ steps.

Washout experiments were carried out following GapR binding and force-linking number-extension experiments by setting DNA tension to approximately 1 pN and then flowing through 200 µL of protein-free GapR assay buffer through the 30 µL flow cell over approximately 1 min, similar to experiments of *Skoko et al., 2004*. Following GapR washout, extension experiments as a function of force and linking number were carried out. All MT data were analyzed using Prism 7.

## Growth rate analysis

Growth rate was measured using a Synergy H1 plate reader. Cells were grown overnight without inducer, diluted, and grown into mid-log $OD_{600}$ 0.2–0.4. Cells were then diluted to $OD_{600}$ 0.01 in 96-well plates in the presence or absence of aTc and grown for 8 hr at 37°C with shaking at 200 rpm.

## Chromatin immunoprecipitation sequencing (ChIP-seq)

GapR and GapR[1-76] ChIP in *E. coli* was performed as previously described for *Caulobacter* (*Guo et al., 2018*). Briefly, cell cultures (20 mL) were grown to $OD_{600}$ ~0.3, diluted back to OD ~0.01, and 25 ng/mL aTc was added for induced cultures. Cells were grown for 2 hr (OD ~0.3) and then fixed by the addition of 10 mM sodium phosphate (pH 7.6) and 1% formaldehyde (final concentrations) (Sigma). When required, 25 µg/mL of rifampicin (Sigma) was added to cells for 20 min prior to fixation. Fixed cells were incubated at RT for 10 min and then quenched with 0.1 M glycine (Sigma) for 5 min at RT followed by 15 min on ice. Cells were washed three times with 1× PBS (pH 7.4) and resuspended in 500 µL of TES buffer (10 mM Tris-HCl [pH 7.5], 1 mM EDTA, 100 mM NaCl), frozen in liquid nitrogen, and stored at −80°C until use. Cells were then thawed and 35,000 U of Ready-Lyse (Epicentre) was added. Following 15 min incubation at RT, 500 µL of ChIP buffer (16.7 mM Tris-HCl [pH 8.1], 167 mM NaCl, 1.1% Triton X-100, 1.2 mM EDTA) containing protease inhibitors (SIGMA*FAST* Protease Inhibitor Tablets) was added. After 10 min at 37°C, the lysates were sonicated on ice and cell debris cleared by centrifugation. Supernatant protein concentration was measured by Bradford assay (Thermo Scientific) and 500 µg of protein were diluted into 1 mL of

ChIP buffer + 0.01% SDS. The diluted supernatants were pre-cleared for 1 hr at 4°C on a rotator with 50 µL of Protein-A Dynabeads (Life Technologies) pre-blocked overnight in ChIP buffer + 0.01% SDS and 100 µg ultrapure BSA (Ambion). Beads were pelleted and 90 µL of the supernatant was removed as input DNA and stored at −80°C, the remaining pre-cleared supernatant was incubated rotating at 4°C overnight with 1 µL of FLAG antibody (Sigma). The immune complexes were captured for 2 hr at 4°C with 50 µL of pre-blocked Protein-A Dynabeads. Beads were then washed consecutively at 4°C for 15 min with 1 mL of the following buffers: low salt wash buffer (0.1% SDS, 1% Triton X-100, 2 mM EDTA, 20 mM Tris-HCl [pH 8.1], 150 mM NaCl), high salt wash buffer (0.1% SDS, 1% Triton X-100, 2 mM EDTA, 20 mM Tris-HCl [pH 8.1], 500 mM NaCl), LiCl wash buffer (0.25 M LiCl, 1% NP-40, 1% deoxycholate, 1 mM EDTA, 10 mM Tris-HCl [pH 8.1]), and twice with TE buffer (10 mM Tris-HCl [pH 8.1], 1 mM EDTA). Complexes were then eluted twice from the beads with 250 µL of freshly prepared elution buffer (1% SDS, 0.1 M NaHCO$_3$). To reverse crosslinking, 300 mM of NaCl and 2 µL of RNase A (0.5 mg/mL) (QIAGEN) were added to the collective eluates, which were incubated at 65°C overnight. Samples were then incubated at 45°C for 2 hr with 5 µL of Proteinase K (NEB) in the presence of 40 mM EDTA (pH 8.0) and 40 mM Tris-HCl (pH 6.8). DNA from the samples was then extracted twice with phenol:chloroform:isoamyl alcohol (25:24:1) (Sigma) and subsequently precipitated by adding 3 M sodium acetate (pH 5.2), 15 µg glycoblue (Ambion) and 1 vol of ice cold isopropanol, and stored at −20°C overnight. DNA was pelleted and washed with 75% ethanol and resuspended in TE buffer (pH 8.0). Input ChIP libraries were generated processing 50 µL of the yeast lysate, by reversing crosslinks, Proteinase K treatment, and phenol:chloroform:isoamyl alcohol extraction as with ChIP DNA.

For *S. cerevisiae*, ChIP was performed as previously described (*Neurohr et al., 2019*) with the following modifications. Cells were grown in YEP + 2% glycerol to OD$_{600}$ ~0.5, and 2% galactose was added for 6 hr while culture OD was maintained between OD 0.5–1.0. For α-factor arrest experiments, cells were grown in YEP + 2% raffinose to OD$_{600}$ ~0.4, arrested in α-factor for 2 hr, before addition of 2% galactose for 6 hr. Cells were then fixed for 15 min at RT with 1% final concentration formaldehyde followed by quenching with glycine. 100 mL of cells were harvested by centrifugation and then washed twice with 1× PBS (pH 7.4), resuspended in 1 mL FA-lysis buffer (50 mM HEPES-KOH pH7.5, 150 mM NaCl, 1 mM EDTA, 1% Triton X-100, 0.1% sodium deoxycholate, 0.1% SDS) and transferred into a 2 mL screw-cap Eppendorf tube. Cells were pelleted quickly in a tabletop centrifuge and the supernatant was discarded by aspiration. Cells were then resuspended in 500 µL FA-lysis buffer, snap-frozen in liquid nitrogen, and stored at −80°C until use. Cells were then thawed, and FA-lysis buffer (containing SIGMA*FAST* Protease Inhibitor Tablets) and SDS was supplemented to bring up the volume to 1 mL with 0.5% SDS final concentration. 1 mL of glass beads were added and the cells were disrupted on a Fast Prep until 80–90% of cells were lysed (intensity 6.5, each 45 s cycle followed by 5 min of cooling, 5–10 cycles in total as confirmed by visual inspection). Cell debris was separated from beads by piercing the tube cap and bottom with a syringe needle, inverting the tube over a 1 mL tip in a 15 mL conical tube, and centrifuged for 1 min at 800 rpm. 9 mL of FA-lysis buffer (with protease inhibitor) was added and the lysate was ultracentrifuged in an SW41 rotor at 32,700 rpm for 20 min to pellet chromatin. The pellet was mechanically disrupted with a wooden stick and transferred to a 1.5 mL TPX microcentrifuge tube (Diagenode), resuspended in 250 µL FA-lysis buffer (with protease inhibitor), and pipetted to resuspend. Samples were sonicated at 4°C in a Bioruptor Waterbath Sonicator (Diagenode) for five cycles (30 s on, 30 s off, high intensity), followed by further pipetting to fully resuspend chromatin. Samples were then sonicated for an additional 3 × 10 cycles (30 s on, 30 s off, high intensity) and quick-spun in a picofuge to recover material from tube walls every 10 cycles. After sonication, 500 µL additional FA buffer was added to the lysate and cellular debris was discarded by centrifugation at 4°C (15 min, ~20,000 × *g*). The supernatant was transferred to a fresh Eppendorf tube, snap-frozen, and stored at −80°C. Total protein was measured by Bradford assay, samples were diluted to contain 1 mg of protein in 1 mL ChIP buffer + 0.01% SDS. Samples were then processed as with bacterial ChIP-seq using 1 µL of anti-FLAG antibody (Sigma) for each immunoprecipitation.

ChIP-seq libraries were built from immunoprecipitated DNA by first end repairing the DNA with 5 µL T4 DNA polymerase (NEB), 5 µL T4 PNK (NEB), and 1 µL Klenow large fragment (NEB) in 100 µL T4 DNA ligase buffer with 0.25 mM dNTPs for 30 min at RT. Repaired DNA was recovered by Ampure XP (Beckman Coulter) bead purification using 100 µL beads in 300 µL 20% PEG/NaCl solution. Beads were washed twice with 80% ethanol, dried, and resuspended in 32 µL EB. The bead

slurry was directly treated with 3 µL Klenow (3′→5′ exo-) (NEB) in 50 µL NEB Buffer #2 with 0.2 mM ATP at 37℃ for 30 min to add 3′ overhangs to DNA. Repaired DNA was recovered by Ampure XP capture and resuspended in 23 µL EB. Y-shaped adaptors were prepared by annealing Illumina PE adapter 1 and Illumina PE adapter 2. Y-shaped adapters were added to bead slurry, and the mix was ligated in 50 µL total volume with 1.5 µL T4 DNA Ligase (NEB) for 1 hr at RT. Ligated library DNA was recovered and free and ligated adapters discarded using an Ampure XP capture at 0.85× volume. Library DNA was recovered by eluting Ampure beads with 33 µL 10 mM Tris-HCl (pH 8.0). DNA libraries were amplified in 50 µL final volume with 2X KAPA HiFi Master Mix (Roche) supplemented with 5% final concentration DMSO (Sigma) and appropriate barcoded primers. The total number of cycles was optimized for each sample to minimize the number of cycles required for library generation. Libraries were purified by two-step Ampure XP capture by first adding 0.5× reaction volume Ampure XP and transferring the supernatant to a fresh tube to discard large fragments, followed by a second capture by adding Ampure XP to 0.82× of the original reaction volume to recover 200–500 bp amplified libraries. DNA was recovered from Ampure beads by resuspending in 20 µL 10 mM Tris-HCl (pH 8.0). Insert size of ChIP libraries was determined to be ~250 bp on average. Paired-end sequencing of libraries was performed on either a NextSeq or a MiSeq at the MIT Bio Micro Center.

*C. crescentus* GapR-3xFLAG ChIP-seq was from GSE100657 (*Guo et al., 2018*). *S. cerevisiae* Scc1 enrichment in the presence of microtubule tension was from GSE104135 (*Paldi et al., 2020*).

## RNA sequencing (RNA-seq)

RNA-seq in *E. coli* was performed as previously described (*Culviner and Laub, 2018*). Cells were harvested immediately before ChIP-seq. 5 mL of cells were harvested by into a 5% phenol, 95% ethanol stop solution. RNA was harvested by phenol/chloroform extraction and treated with 2 µL Turbo DNase (Invitrogen) with 5 µL SuperaseIN (Invitrogen) in 100 µL total volume at 37℃ for 20 min. RNAs were subsequently recovered by phenol/chloroform extraction. rDNA was removed using in-house protocols (*Culviner et al., 2020*). mRNA was subsequently fragmented with RNA fragmentation reagents (Invitrogen) and cDNA was generated with random primers and Superscript III (Invitrogen). Second-strand synthesis was conducted using dUTP instead of dTTP and RNase H (NEB), *E. coli* DNA ligase (NEB), and DNA Pol I (NEB) were added, followed by an incubation at 16℃ for 2.5 hr. cDNA was recovered by Ampure XP (Beckman Coulter) bead purification. cDNA was then end-repaired and converted into libraries as for ChIP-seq samples. Before library amplification, the dUTP-containing second strand was digested by adding 1 µL of USER enzyme (NEB) and incubating at 37℃ for 15 min, followed by 95℃ for 5 min to inactivate USER enzyme. Libraries were generated as for ChIP-seq samples, and paired-end sequencing was performed on a NextSeq at the MIT Bio Micro Center.

RNA-seq in *S. cerevisiae* was performed by harvesting cells immediately before ChIP-seq or in mid-log phase ($OD_{600} < 1.0$). 5–10 mL of cells were pelleted at 8000 rpm for 5 min and washed with 500 µL $H_2O$ before being snap-frozen in liquid nitrogen. RNA was isolated by resuspension in 500 µL 10 mM Tris-HCl (pH 7.5), 10 mM EDTA, and 0.5% SDS followed by addition of 500 µL hot acid-phenol. Cells were then shaken in a thermomixer at 65℃ at 2000 rpm for 20 min, before incubation for 5 min on ice, and phenol extraction. RNA was then isolated by phenol-chloroform extraction followed by a chloroform wash, and precipitation with isopropanol. gDNA was removed by addition of Turbo DNase and total RNA was subsequently recovered by phenol/chloroform extraction. mRNA was isolated using poly(dT) pulldown using the NEBNext Poly(A) mRNA Magnetic Isolation Module (NEB) and processed into libraries as with *E. coli* RNA. Paired-end sequencing was performed on a NextSeq at the MIT Bio Micro Center.

## Electrophoretic mobility shift assays (EMSA)

For EMSA, linear 210 bp DNA was generated by PCR and purified with PCR Purification Kits (QIAGEN). Reactions (15 mL) with indicated amounts of GapR and 210 bp DNA (40 ng) in binding buffer (40 mM sodium phosphate [pH 7.5], 100 mM NaCl, 50 mg/mL ultrapure BSA, 0.5 mM DTT) were incubated at 30℃ for 60 min and then placed on ice. DNA loading buffer was added and 10 mL of the reactions were electrophoresed on 6% DNA Retardation gels (Invitrogen) at 130 V for 60 min at

4˚C. Gels were stained in SYBR Gold (Invitrogen) and imaged with a Typhoon FLA 9500 imager (GE Lifesciences).

## Sequencing data processing

Data analysis was performed with custom scripts in Python 3.6.9. For all histograms, a kernel density estimation (KDE) was generated, with the y-axis units indicating KDE density. All t-tests performed were two-tailed tests.

For *E. coli* ChIP, paired-end reads were mapped to the MG1655 reference genome (NC_000913.2) using bowtie2 with default parameters (*Langmead and Salzberg, 2012*). For *S. cerevisiae* ChIP-seq, paired-end reads were mapped to the reference genome (S288C Scer3) using bowtie2 with default parameters (*Langmead and Salzberg, 2012*). Once aligned, unique reads were isolated and read extension and pile-up was performed using the bedtools function genomeCoverageBed (*Quinlan and Hall, 2010*) and converted into wig format using custom Python scripts. The data were then smoothed by convolution with a Gaussian (m = 0, s = 250 bp, x = (1000 bp, +1000 bp)) and then normalized to reads per million (rpm). Because different growth conditions (e.g., with and without α-factor arrest) led to variable rDNA copy number, experiments were normalized to total count excluding chromosome XII (containing the rDNA). Data were then smoothed over 250 bp. To generate *S. cerevisiae* GapR enrichment, a pseudocount was added to each position and the GapR-3xFLAG ChIP was normalized by the GapR$^{WT}$ ChIP (GapR-3xFLAG ChIP + 0.01)/(GapR$^{WT}$ ChIP + 0.01). Scc1 ChIP-seq occupancy ratio was calculated from GSE104135 as reported (*Paldi et al., 2020*). Correlation between two ChIP experiments was generated by binning data every 100 nt.

For *E. coli* RNA-seq, paired-end reads were mapped to the MG1655 reference genome (NC_000913.2) and to the GapR expression plasmid using bowtie2 with default parameters (*Langmead and Salzberg, 2012*). Duplicated reads were filtered out and the read coverage was mapped genome by assigning each mapped base a value of 1. To calculate mRNA expression levels, the number of reads mapped to a gene was divided by the length of the gene and normalized to yield the mean number of reads per kilobase of transcript per million sequencing reads (rpkm).

To determine if ectopic expression of GapR alters global supercoiling in *E. coli*, we compared the rpkm of all expressed genes with and without GapR expression (genes with rpkm > 20 in either condition, ~2500 genes). To examine the effects of GapR on expression of known supercoiling-sensitive genes, we compared the rpkm of genes known to be induced or repressed upon topoisomerase inhibition (*Peter et al., 2004*) as well as for the DNA gyrase and topo IV subunits P$_{gyrA}$, P$_{gyrB}$, P$_{parC}$, and P$_{parE}$, which have been reported to be supercoiling-sensitive in *E. coli* or other bacteria (*Ferrándiz et al., 2016*; *Menzel and Gellert, 1987*).

*S. cerevisiae* RNA-seq was analyzed by aligning to SacCer3 using bowtie2 with default parameters (*Langmead and Salzberg, 2012*). Duplicated reads were filtered out and the read coverage and rpkm values were calculated as for *E. coli*. To determine if ectopic expression of GapR alters global supercoiling or activates stress responses, we examined the change in expression with and without GapR of genes known to be supercoiling sensitive (*Pedersen et al., 2012*) or that are transcriptionally activated by stress-responsive signaling pathways.

## Identifying AT-bias and GapR-associated DNA motifs

AT content at each base pair was computed using a centered 100 bp sliding window. To identify correlations between AT content and GapR binding, AT content was plotted versus GapR ChIP at each position. To identify DNA sequence motifs enriched in GapR-bound sequences, we isolated the 35 regions with highest GapR ChIP signal as was described previously in *C. crescentus* (*Guo et al., 2018*). For *E. coli* ChIP, we isolated regions above 0.843 rpm (regions less than 150 bp apart were merged) as input sequences and regions below 0.30 rpm for control sequences. For *S. cerevisiae*, we isolated GapR ChIP regions above 0.376 rpm from smoothed data (regions less than 150 bp apart were merged) and regions below 0.055 rpm for control sequences. A 200 bp window centered at the maximum (or minimum, for control sequences) signal intensity of each of these regions was retrieved and submitted to DREME for sequence motif analysis (*Bailey, 2011*). The highest DREME motif is reported.

### Assessing GapR 5' and 3' end enrichment

For *E. coli*, TU annotation was taken from the Ecocyc operon annotation. GapR occupancy at the 5' and 3' ends were calculated by examining TUs $\geq$ 1500 bp and determining the change in GapR per base in the 1000 bp before and after the transcriptional start site or the transcription termination site: for example, mean(GapR$_{-1000..start}$) − mean(GapR$_{start..1000}$). TUs were filtered to prevent redundancy; for divergently or convergently transcribed regions that are within 1000 bp, GapR occupancy is only calculated for one strand.

For *S. cerevisiae*, publicly available annotation datasets do not contain transcriptional start or termination site information, and only include coordinates for the coding regions of genes. Gene start and end positions were used as a proxy for transcriptional start and termination sites. GapR occupancy was determined by examining genes $\geq$ 1000 bp and calculating the mean normalized GapR enrichment in the 500 bp before or after the start or end of genes: for example, mean(GapR$_{end...500}$). TUs were filtered to prevent redundancy; for divergently or convergently transcribed regions that are within 500 bp, GapR occupancy was only calculated for one strand.

### Assessing transcription-dependent GapR 5' and 3' end enrichment

For *E. coli*, GapR occupancy at the 5' and 3' ends were calculated by examining TUs $\geq$ 1500 bp and determining the mean change in GapR in the 1000 bp before and after the transcriptional start site or the transcription termination site: for example, (mean(GapR$_{-1000..start}$) − (mean(GapR$_{start..1000}$))). Transcription-dependent change in GapR occupancy at the 5' and 3' ends was calculated by examining TUs $\geq$ 1500 bp and determining the mean change in GapR in the presence and absence of rifampicin in the 1000 bp before and after the transcriptional start site or the transcription termination site: for example, (mean(GapR$_{-1000..start}$) − mean(GapR + Rif$_{-1000..start}$)) − (mean(GapR$_{start..1000}$) − mean (GapR + Rif$_{start..1000}$)). The transcriptional strength was calculated for each TU from GapR-3xFLAG induced RNA-seq data by determining the mean number of reads mapped over each TU and normalizing to yield the mean number of rpkm. TU rpkm cutoffs were chosen to isolate the highest expressing 125 and 250 TUs (>65, >25.7 rpkm), and the lowest expressing 250 TUs (<3.284 rpkm). TUs were filtered to prevent redundancy; for divergently or convergently transcribed regions that are within 1000 bp, GapR occupancy is only calculated for one strand. To generate heatmaps of GapR enrichment at 5' and 3' ends of genes, TUs $\geq$ 1500 bp were sorted by expression level and the change in GapR in the presence and absence of rifampicin in 6 kb window around the transcriptional start site or the transcription termination site (e.g., GapR$_{-4000..start..2000}$ − GapR + Rif$_{-4000..start..2000}$) was plotted for the 300 highest and lowest expression TUs.

For *S. cerevisiae*, transcriptional strength was calculated similarly, except by examining TUs $\geq$ 1000 bp and determining the mean GapR enrichment in the 500 bp before or after genes. Transcriptional cutoffs were chosen to isolate the highest expressing 125, 250, and 500 genes (> 455, >225.3, >110 rpkm) and the lowest expressing 500 genes (< 9.385). TUs were filtered to prevent redundancy; for divergently or convergently transcribed regions that are within 500 bp, GapR occupancy is only calculated for one strand.

### Identifying GapR-enriched regions

For *E. coli* GapR peaks, we isolated the top 5% of positions with greatest transcription-dependent GapR enrichment (GapR$_i$ − GapR + Rif$_i$ $\geq$ 0.118). The borders of each GapR-bound region surrounding the enrichment peak were identified by determining where transcription-dependent GapR enrichment was above the mean + 1/3 of a standard deviation. Regions less than 150 bp apart are then merged.

For *S. cerevisiae* GapR peaks, we isolated the top 5% of positions with greatest GapR enrichment (e.g., GapR$_i$ $\geq$ 1.657 in raffinose). The borders of each GapR-bound region surrounding the enrichment peak were identified by determining where transcription-dependent GapR enrichment was above the mean + 1/3 of a standard deviation. Regions less than 150 bp apart are then merged.

### Assessing transcription orientation-dependent GapR enrichment

For *E. coli*, highly expressed TUs ($\geq$ 17.3 rpkm, top third) were analyzed. TUs below the rpkm cutoff were discarded and assumed to be transcriptionally silent. For the remaining TUs, the regions between TUs were categorized based on whether the downstream TU is convergent, divergent, or

in the same orientation. Intragenic regions < 50 bp were removed from the analysis. Mean GapR ChIP or mean transcription-dependent GapR ChIP was then calculated for each region and within TUs. The same analysis was repeated for determining the mean GapR[1-76] ChIP.

To examine transcription orientation at GapR ChIP peaks, peaks were identified as reported above. The same number of unenriched regions was identified by isolating the 7% of positions with lowest transcription-dependent GapR enrichment ($GapR_i$ – GapR + $Rif_i \leq -0.0885$). The borders of each GapR-bound region surrounding the enrichment peak were identified by determining where transcription-dependent GapR enrichment was below the mean $-1/3$ of a standard deviation ($GapR_i$ – GapR + $Rif_i < -0.024$). Regions less than 150 bp apart are then merged. At each GapR-enriched or unenriched region, the transcriptional propensity of the surrounding area was determined by the following procedure. First, the mean number of reads on the forward and reverse strands was calculated for each region $\pm$ 5 kb on both sides; if the mean reads < 0.01, the region is assumed to be silent (fwd + rev < 0.01 = silent). Next, the midpoint of the enriched/unenriched region was determined, and the transcriptional strength for each strand was calculated from the midpoint to 2 kb past either end of the region. Transcriptional propensity is then assigned based on the relative transcriptional strength: $fwd_{left}$ > $rev_{left}$ and $rev_{right}$ > $fwd_{right}$ = convergent; $fwd_{left}$ < $rev_{left}$ and $rev_{right}$ < $fwd_{right}$ = divergent; other cases are assumed to be the same orientation. If the mean reads is < 0.01 within the 2 kb window, the window was expanded to 5 kb and the analysis repeated for the orientation assignment. Fisher's exact test was used to determine if GapR-enriched regions were more frequently associated with convergent transcription and de-associated with divergent transcription.

For *S. cerevisiae*, all genes were analyzed. Regions between genes were categorized based on whether the downstream gene is convergent, divergent, or in the same orientation. Mean GapR ChIP was then calculated for each region and within genes. Intragenic regions < 50 bp were removed from the analysis. To determine the transcriptional orientation at GapR-enriched and -unenriched regions, peaks were identified as detailed above. GapR-unenriched were the 5% of positions with least GapR enrichment ($GapR_i \leq 0.626$). The borders of each GapR-bound region surrounding the enrichment peak were identified by determining where transcription-dependent GapR enrichment was above the mean + $1/3$ of a standard deviation. Regions less than 150 bp apart are then merged. At each GapR-enriched or -unenriched region, the transcriptional propensity of the surrounding area was determined by the same procedure as with *E. coli* regions, except examining GapR ChIP.

## Correlation between GapR and upstream or downstream transcription

To determine if intergenic GapR is associated with upstream or downstream transcription, the mean GapR enrichment between all co-directionally organized *S. cerevisiae* gene pairs with an intergenic distance > 50 bp was calculated. Mean GapR enrichment was then correlated (Pearson's correlation coefficient) with the transcription level of either the upstream or downstream gene. To isolate poorly transcribed downstream genes, a rpkm cutoff of < 20 was used, which yielded 1207 gene pairs.

## MNase-seq and DNase-seq data processing

*S. cerevisiae* MNase-seq from GSM3069971 (*Cutler et al., 2018*) was analyzed by aligning to SacCer3 using bowtie2 with default parameters (*Langmead and Salzberg, 2012*). Reads were sorted and filtered with samtools (*Li et al., 2009*), and the center of each paired read was interpreted as the nucleosome dyad and plotted, as reported previously (*Cutler et al., 2018*).

*S. cerevisiae* DNase-seq from GSM1705337 (*Zhong et al., 2016*) was analyzed by aligning to SacCer3 using bowtie with the following parameters (*Langmead et al., 2009*) to map the first 20 base pairs for each read: -n 2 l 20–3 30 m 1 `-best -strata`. The position at first base pair (5' end) of the alignment was assigned as the DNase cleavage site and given a mapped value of 1 and the total number of DNase reads were tabulated separately for the forward and reverse strands, as reported previously (*Zhong et al., 2016*). For some analyses, the DNase-seq coverage was transformed by a $\log_{10}$ transform after addition of a pseudocount to each base: $\log_{10}(DNase\text{-}seq + 1)$.

## MNase-seq and DNase-seq data analysis

To examine nucleosome occupancy and DNase hypersensitivity at GapR-enriched regions, first GapR-enriched regions were identified as detailed above. The regions were centered around the

position of maximum GapR occupancy, and the mean GapR enrichment, MNase-seq, and DNase-seq at each base was determined for 1000 bp around the GapR peak. Correlation between GapR ChIP and DNase-seq or MNase-seq experiments was generated by binning data every 100 nt.

To identify open chromatin from DNase-seq, we isolated the top ~5% of positions with greatest DNase-seq reads (sum of the forward and reverse strand reads, DNase-seq$_i \geq$ 12). The borders surrounding each DNase-hypersensitive region were determined by where DNase-seq was above the mean + 1/3 of a standard deviation (DNase-seq$_i >$ 12.3). Regions less than 10 bp apart are merged. To calculate GapR enrichment in open regions, the mean log$_2$(GapR enrichment) was calculated for all DNase-seq peaks longer than 50 bp (8005 unique regions). This 50 bp length cutoff was used to ensure that the absence of GapR binding was not due to regions being shorter than a GapR binding site. For heatmaps, the maxima of the top 500 DNase-seq or GapR peaks was used as the midpoint, and the GapR, DNase-seq, or MNase-seq in a 4 kb window surrounding the peak is shown, with 10 bp binning.

To compare the AT content of GapR-enriched and DNAse-accessible regions, the top 500 GapR-enriched regions and the top 500 DNase-accessible regions longer than 50 bp were examined. The AT content was determined for the 10 bp surrounding the GapR-seq or DNase-seq maxima of each region.

To assess MNase-seq and DNase-seq at 5' and 3' ends, genes $\geq$ 1000 bp were examined and the mean MNase-seq and log$_{10}$(DNase-seq + 1) in the 500 bp before or after the start or end of genes was calculated: for example, mean(MNase-seq$_{end...500}$). TUs were filtered to prevent redundancy; for divergently or convergently transcribed regions that are within 500 bp, mean MNase-seq and DNase-seq was calculated for only one strand.

To assess MNase-seq and DNase-seq reads based on transcription orientation, datasets were analyzed as reported above with GapR-seq enrichment. Briefly, all regions between genes were categorized based on whether the downstream gene is convergent, divergent, or in the same orientation. Mean MNase-seq and log$_{10}$(DNase-seq + 1) was calculated for each region and within genes. Intragenic regions < 50 bp were removed from the analysis in order to examine regions that would be accessible by GapR.

## Comparison of psoralen tiling array and GapR-seq

*S. cerevisiae* psoralen enrichment from GSE114410 (*Achar et al., 2020*) was analyzed by downloading the BedGraph file containing the psoralen enrichment score (bTMP IP/input) for short interval bases of wild-type cells in G1 phase grown in glucose at 28°C (GSM3141352). Psoralen enrichment score was plotted compared to GapR enrichment, and correlation plots between GapR ChIP and psoralen score were generated by binning data by 100 nt.

## Centromere, pericentromere, and cohesin analysis

The mean GapR enrichment at all centromeres was determined and compared to the mean GapR enrichment at all intergenic regions. Centromeres were aligned by their left position (oriented CDEI-CDEII-CDEIII), and the mean GapR enrichment and 95% confidence interval at each base at all centromeres were determined for 4120 bp around the centromere.

Borders of pericentromeres were defined based on published analysis of cohesin-binding and convergent genes (*Paldi et al., 2020*). Scc1 ChIP-seq occupancy ratio in the presence of microtubule tension was taken from GSE104135 (*Paldi et al., 2020*). To identify Scc1 peaks, we isolated the top 500 regions with greatest Scc1 enrichment first in the presence and then in the absence of tension. For heatmaps, the maxima of the top 500 Scc1 or the GapR peaks was used as the midpoint, and the GapR or Scc1 enrichment in a 4 kb window surrounding the peak is displayed, with 10 bp binning.

## ARS analysis

The mean GapR enrichment at all ARS was determined and compared to the mean GapR enrichment at all intergenic regions. To generate heatmap of GapR enrichment at ARS, all ARS were aligned by their left position and the GapR enrichment was determined for a window −1000 to +2000 bp with 10 bp binning from this position.

## S1-DRIP-seq analysis

*S. cerevisiae* S1-DRIP-seq from SRP071346 (*Wahba et al., 2016*) was analyzed by aligning to SacCer3 using bowtie2 with default parameters. Genome coverage was mapped with bedtools function genomeCoverageBed and converted into wig format using custom Python scripts (*Quinlan and Hall, 2010*). Transposable (Ty) element locations were taken from Saccharomyces Genome Database. To generate alignment profiles of GapR at all Ty elements and accommodate the fact that Ty elements vary in length, each Ty element was divided into 20 bins, with the middle bin being larger or smaller to accommodate the overall size. 10 more equivalently sized bins were then extended to either side of the Ty element (~2500 bp). The mean GapR enrichment and S1-DRIP-seq with 95% confidence intervals were then determined for all bins. At telomeres, the mean GapR enrichment and S1-DRIP-seq with 95% confidence intervals were determined for the 1500 bp divided into 50 bp bins flanking each telomere. For *Figure 7—figure supplement 1F*, GapR enrichment and S1-DRIP-seq were determined for 1500 bp divided into 50 bins starting from the first nucleotide (towards CEN) after the telomeric repeat sequence.

## Data and code availability

Datasets generated during this study are deposited at the Gene Expression Omnibus (GEO): GSE152882 (https://www.ncbi.nlm.nih.gov/geo/query/acc.cgi?acc=GSE152882). All custom-made scripts used in this study are available in the GitHub repository (https://github.com/msguo11/GapR_seq_analysis; *Guo, 2021*; copy archived at swh:1:rev:cb9b4e053a4160bd380aecf9f0cf2d18b4c708b7).

## Acknowledgements

We thank M Leroux, P Culviner, D Haakonsen, C Tsokos, O Rando, J Gerton, and A Maxwell for comments on the manuscript. We thank S Srikant, M Looke, and G Neurohr for strains and vital technical assistance, and members of the Laub lab for critical feedback.

## Additional information

### Competing interests

Michael T Laub: Reviewing editor, *eLife*. The other authors declare that no competing interests exist.

### Funding

| Funder | Grant reference number | Author |
| --- | --- | --- |
| National Institutes of Health | K99GM134153 | Monica S Guo |
| National Institutes of Health | U54CA193419 | John F Marko |
| National Institutes of Health | U54DK107980 | John F Marko |
| National Institutes of Health | R01GM105847 | John F Marko |
| National Institutes of Health | R01GM082899 | Michael T Laub |
| National Institutes of Health | S10OD026741 | Monica S Guo |
| Howard Hughes Medical Institute | Investigator | Michael T Laub |
| National Institutes of Health | UM1-HG011536 | John F Marko |

The funders had no role in study design, data collection and interpretation, or the decision to submit the work for publication.

### Author contributions

Monica S Guo, Conceptualization, Resources, Data curation, Formal analysis, Funding acquisition, Validation, Investigation, Visualization, Methodology, Writing - original draft, Writing - review and editing; Ryo Kawamura, Conceptualization, Resources, Data curation, Formal analysis, Investigation,

Methodology; Megan L Littlehale, Formal analysis, Investigation; John F Marko, Conceptualization, Data curation, Formal analysis, Writing - review and editing; Michael T Laub, Conceptualization, Formal analysis, Supervision, Funding acquisition, Visualization, Writing - original draft, Project administration, Writing - review and editing

#### Author ORCIDs
Monica S Guo ![ORCID] https://orcid.org/0000-0002-6720-2323
John F Marko ![ORCID] http://orcid.org/0000-0003-4151-9530
Michael T Laub ![ORCID] https://orcid.org/0000-0002-8288-7607

#### Decision letter and Author response
Decision letter https://doi.org/10.7554/eLife.67236.sa1
Author response https://doi.org/10.7554/eLife.67236.sa2

# Additional files

### Supplementary files
• Transparent reporting form

### Data availability
Datasets generated during this study are deposited at the Gene Expression Omnibus (GEO): GSE152882.

The following dataset was generated:

| Author(s) | Year | Dataset title | Dataset URL | Database and Identifier |
|---|---|---|---|---|
| Guo MS, Laub MT | 2021 | High-resolution, genome-wide mapping of positive supercoiling in chromosomes | https://www.ncbi.nlm.nih.gov/geo/query/acc.cgi?acc=GSE152882 | NCBI Gene Expression Omnibus, GSE152882 |

The following previously published datasets were used:

| Author(s) | Year | Dataset title | Dataset URL | Database and Identifier |
|---|---|---|---|---|
| Guo MS | 2018 | GapR-3xFLAG ChIP-seq | https://www.ncbi.nlm.nih.gov/geo/query/acc.cgi?acc=GSM2690550 | NCBI Gene Expression Omnibus, GSM2690550 |
| Robertson D | 2020 | Scc1-6HA WT Tension IP | https://www.ncbi.nlm.nih.gov/geo/query/acc.cgi?acc=GSM3668254 | NCBI Gene Expression Omnibus, GSM3668254 |
| Cutler S | 2018 | MNase_YPD30_WT-B_166_40U.sgr.gz | https://www.ncbi.nlm.nih.gov/geo/query/acc.cgi?acc=GSM3069971 | NCBI Gene Expression Omnibus, GSM3069971 |
| Belsky JA | 2016 | DNase-seq_W303_S_cerevisiae_Sample1 | https://www.ncbi.nlm.nih.gov/geo/query/acc.cgi?acc=GSM1705337 | NCBI Gene Expression Omnibus, GSM1705337 |
| Iqbal MAM | 2020 | WT_Supercoil_G1_28_IP | https://www.ncbi.nlm.nih.gov/geo/query/acc.cgi?acc=GSM3141352 | NCBI Gene Expression Omnibus, GSM3141352 |
| Wahba L, Costantino L, Tan FJ, Zimmer A, Koshland D | 2016 | wildtype-rep1 | https://trace.ddbj.nig.ac.jp/DRASearch/experiment?acc=SRX1761639 | DDBJ Sequence Read Archive, SRR3504389 |

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
