## [Decision Letter]

**Acceptance summary:**

DNA supercoiling plays a key role in the regulation of DNA replication and transcription. Both processes affect DNA supercoiling locally; thus, the precise distribution of DNA superhelicity is expected to be highly dynamic and change depending on cellular status. Whether there exist chromosomal regions that display preferential supercoiling levels has been unknown due to a paucity of technologies for measuring supercoiling throughout the genome. The generation and removal of topologically stressed DNA is required for cell viability but, when mishandled, can lead to molecular pathologies. The work described in the present manuscript uses GapR, a protein that preferentially binds overwound DNA, to map genomic regions of positive supercoiling in bacteria and yeast. Such insights are needed to understand how supercoiling is partitioned and controlled at chromosome-wide level.

**Decision letter after peer review:**

Thank you for submitting your article "High-resolution, genome-wide mapping of positive supercoiling in chromosomes" for consideration by *eLife*. Your article has been reviewed by 2 peer reviewers, and the evaluation has been overseen by a Reviewing Editor and Naama Barkai as the Senior Editor. The reviewers have opted to remain anonymous.

Essential revisions:

Please see the request for revisions as detailed below.

*Reviewer #1:*

In this study, the authors present a new method that is claimed to measure (+) supercoiling levels in vivo.

The topology of chromosomal DNA in bacteria is known to be highly dynamic and to change from cell to cell. The method proposed by the authors would thus detect regions with 'persistent' or 'preferential' levels of (+) DNA supercoiling. Such a measurement would be relevant to understanding bacterial transcription, replication and DNA segregation.

In a previous study, the authors show that GapR binds positively supercoiled DNA in vivo using biochemical methods (Guo. et al., Cell 2018). In the present study they further investigate the mechanism of (+) supercoiled DNA binding using magnetic tweezers.

Their first experiment shows that, in the presence of GapR, introduction of (+) supercoiling reduces the extension of the DNA molecule but to a lower degree than in absence of GapR. The authors conclude that GapR precludes the conversion of (+) supercoiling to (+) writhe. It is not clear though where is the excess (+) supercoiling going. Does GapR favour conversion of the added (+) supercoiling into (+) twist? Is there any clue of this in their crystallographic data? What is their model for how GapR is constraining the added (+) supercoiling? The scheme in Figure 1G is very confusing in this regard.

The authors then monitor changes in DNA extension over time in absence and presence of GapR for (+) and (-) supercoiled DNA substrates. They conclude that GapR dynamically binds and diffuses along (-) supercoiled DNA but stably binds to (+) supercoiled DNA. I do not see the evidence for the former claim (dynamic binding and diffusion/sliding). I do not fully agree with the evidence shown for the second conclusion: 1 μm is a pretty high concentration for magnetic tweezers experiments. What is the binding affinity of GapR? Their moderate effect of GapR at lower concentrations (100nM) suggests that the binding affinity is likely in the hundreds of micromolars. In this range, the residence time of a protein on its substrate is short (sec) thus I am confused as to why the authors say GapR binds stably to (+) supercoiled DNA.

All in all, magnetic tweezers experiments are interesting, but do not clearly demonstrate binding of GapR to (+) supercoiling DNA, do not provide further mechanistic insight or strong support for their model of how GapR binds to (+)/(-) supercoiled DNA. Perhaps a higher resolution assay (e.g rotor bead) could shed more light into the mechanism of binding of GapR.

Next, the authors use ChIP-seq on GapR and a fusion of GapR to 3xFLAG. They see that these proteins bind to AT-rich regions located downstream from a ribosomal operon, and their binding is reduced when transcription is inhibited. In the example shown, there are several highly transcribed ribosomal genes in the operon. I would have expected to see binding of GapR throughout the operon or at least at the 3' ends of these genes, as RNApol2 is producing (+) supercoils throughout the entire operon. However, binding of GapR appears only at the 3' end of the operon. Do these experiments really show that GapR binds to (+) supercoils in *E. coli* as claimed by the authors?

The authors then used mutants of GapR to determine if binding of GapR to DNA was due to chromosome accessibility or to DNA topology. For this they used a deletion mutant of GapR that binds DNA but does not encircle it. They show that this mutant binds (+) supercoiled DNA in a biochemical topological assay. However, it would be more consistent to use the magnetic tweezers assay of Figure 1. Does this mutant also suppress accumulation of (+) writhe? how are the dynamics of binding to (-), (+) DNA affected? These experiments could provide insight and support for their model of binding of GapR to (+) supercoiled DNA (see above).

Next, they used ChIP-seq to show that the deletion mutant fails to bind to AT-rich regions at the 3' ends of highly expressed operons. From this, they conclude that binding to (+) supercoiled regions requires tetramerization. I am not convinced that the data supports this conclusion. If the tetramerization domain was required to increase the affinity of DNA binding of GapR dimers, then one would also expect a loss in specific binding in absence of the tetramerization domain. If this domain was needed for interactions with other proteins present at 3'-ends of highly transcribed genes, then one would also expect a similar result.

*Reviewer #2:*

This work claims to provide a new tool that specifically detects positive DNA supercoiling, genome-wide. The research question asked in the manuscript is quite important. The tool is based on the ability of bacterial protein GapR to bind preferentially with over-twisted DNA. The manuscript is divided into two parts: first, a demonstration that bacterial protein GapR binds preferentially with over-twisted DNA, both in single-molecule assays and in vivo at sites of already predicted to be positively supercoiled, and then a genome-wide search for positive supercoiling at the key genomic positions. There are several promising results in the second part of the manuscript, but they are all based on the first part where the presented evidence/data are not sufficient to draw decisive conclusions. Many additional experiments, much more data and further controls are required to prove that GapR could be used as a probe for positive DNA supercoiling.

1) The authors in their previous work used analysis of the DNA supercoiling induced by plasmid-GapR interaction to suggest that the protein likely binds over-twisted DNA. In the current manuscript, Guo et al., perform again this analysis (Figure 3A, S3E). The weak point in their DNA supercoiling assay is that DNA topology does not change up to 2.5 μM GapR, and then an abrupt change is observed as the concentration is increased. This is not the expected pattern if GapR binding increases incrementally as it introduces or stabilizes a small amount of positive twist in the DNA (see Clark and Leblanc, Methods Mol Biol. 2015 for a recent review of this method). The distribution of topoisomers in the assay should gradually shift from the relaxed state to the new supercoiled position until binding is saturated. One of the simplest explanations of the observed unusual pattern is a synergy threshold: for example, the GapR-mediated DNA bridging which is reported in the literature (see Lourenço et al., mBio. 2020) might give this result. To confirm their suggestion, the authors use single-molecule assays. Based on the pattern of DNA "rotation-extension curve" generated by this assay, the authors state that GapR stably interacts with positively supercoiled DNA while the interaction with negatively supercoiled DNA is unstable. However, as admitted in the manuscript this curve is highly unusual and cannot be explained solely by the constraining of positive supercoils. The high fluctuations of DNA length on the negative-supercoiling side of the curve once again suggests that some kind of cooperative binding-unbinding of GapR affects the shape of the DNA.

2) The authors imply that the GapR-binding method might be superior to psoralen-crosslinking methods for the detection of positive supercoiling but there is no actual comparison. Psoralen assays have been calibrated both in vitro and in vivo (see Bermúdez et al., Nucleic Acids Res. 2010, and Kouzine et al., Nat Struct Mol Biol. 2008). Similar calibration is required for GapR study. In the current manuscript, the authors detected GapR binding at sites expected to be positive supercoiling (Figure 2, 3, and 4) which is not sufficient to support the key claims in the manuscript – GapR is binding at positively supercoiled sites. Based on the known topological plasticity of chromatin to the DNA over-twisting, one might expect that GapR is able to differentiate between difference positive torsional stress stored in twist with that in the writhe of the 3-D shape of DNA in the *E. coli* genome or in yeast chromatin. Although exciting, the full characterization of DNA-GapR interaction is required.

3) The strength of the manuscript is the technically impressive analysis of the GapR localization in the genome reported in the second part of the manuscript. The authors find that this protein does recognize the strategic regions of the genome (Figure 6 and 7). With proper analysis of DNA-GapR interaction in the first part of the manuscript, these data will indicate that GapR is an important probe for DNA conformation in the context of key genomic processes.

1) The quality of the 2D gels should be improved and accurate titration with broader range of protein concentrations should be performed. The explanation of the topoisomers' distribution should minimize pre-assumptions. The ability of the protein (protein preparation) to induce DNA double-stranded breaks and nicks should be explained.

2) The DNA topology electrophoresis and single-molecular assays were performed at different protein concentrations. What is the reason for choosing different concentrations? What would we see if single-molecular assay would be performed at higher protein concentration? All anomalies on the extension curve should be explained (might be added to the Supplementary section). What is the reason for the high fluctuation on the negative side of the curve? I do not think it can be explained by single binding-unbinding-diffusing event. Why is naked, relaxed DNA is shorter than the same DNA in the presence of the protein? One might expect the opposite if GapR constrains ower-twisted DNA.

3) Preferential binding of GapR to positively supercoiling DNA over negatively supercoiled DNA was not proven as the single-molecular assay did not give a definitive answer. The study should be supplemented by competition assay between different DNA conformations. Could you efficiently fish out positively supercoiled plasmid from the mixture of genomic DNA circles/plasmids wound to different degrees? What happens if you compare normal plasmids (able to form supercoils) with DNA minicircles (unable to form supercoils)?

4) In the omics study, GapR binding should be compared with psoralen-based maps. The assumption that psoralen-based studies infer the presence of positive supercoils by the absence of crosslinking is wrong. In the classical approach, the presence of supercoiling is inferred from the changes of psoralen intercalation after fast nicking and relaxation of the DNA inside the cells (Sinden's studies). In addition, it is incorrect to say that "psoralen-based studies are still limited in resolution". With developing of high throughput sequencing methods, the resolution of supercoiling is close to the DNA persistence length (Henikoff's studies).

5) All discussion of the positive supercoiling in yeast should be supplemented with the introduction of known torsional plasticity of chromatin to DNA over-twisting. The current consensus in the field is that chromatin fiber is torsionally soft with respect to positive supercoiling – twisting of chromatin results in the reorganization of nucleosomal template without introducing DNA over-twisting. How do you align this expected topological plasticity with the proposed ability of GapR to sense twist rather than writhe?

6) There is very little discussion on consequences of the prolonged expression of GapR protein. So, one caveat is that the expression of GapR over time does not perturb the DNA topology or chromatin conformation in cells. There may be quick and more general approaches for this method than building strains and cell-lines to express GapR. For example, the authors could make yeast spheroplasts, treat with saponin/digitonin, in the presence of GapR, fix with formalin and then perform ChIP-seq, (this is the sort manipulation that is used in native ChIP) and then compare these results with in vivo expression of GapR. This would potentially eliminate artefacts of prolonged in vivo expression and greatly expand the general utility of the method as it could be than used on any cell line without transgenic or knock-in expression.

7) The sentence "Genomic DNA can become supercoiled when the DNA duplex winds about its own axis forming a right-handed superhelix (positive supercoiling) or a left-handed superhelix (negative supercoiling)" is misleading. In the plectonemic form of unconstrained supercoiling, a right-handed superhelix is assigned a negative number (negative supercoiling) and a left-handed superhelix is assigned a positive number (positive supercoiling). Opposite for solenoid/toriodal form of constrained supercoiling.

8) Moderate binding is detected to 5' ends of weakly expressed genes but not to highly expressed gene. It would be good to check if this is due to an upstream co-directional transcript that pumps positive supercoils into the promoter of the weakly expressed genes making the promoter harder to melt and transcribe.

9) DNase likes G-C rich DNA better than A-T-rich, GapR likes the opposite. Does the base composition partly explain separation of GapR and DNaseI sites?

10) The idea of involvement of positive supercoiling in R-loop genesis should be discussed together with recent work from Chedin' lab (Stolz et al., Proc Natl Acad Sci U S A. 2019).

---

## [Author Response]

Reviewer #1:In this study, the authors present a new method that is claimed to measure (+) supercoiling levels in vivo.The topology of chromosomal DNA in bacteria is known to be highly dynamic and to change from cell to cell. The method proposed by the authors would thus detect regions with 'persistent' or 'preferential' levels of (+) DNA supercoiling. Such a measurement would be relevant to understanding bacterial transcription, replication and DNA segregation.In a previous study, the authors show that GapR binds positively supercoiled DNA in vivo using biochemical methods (Guo. et al., Cell 2018). In the present study they further investigate the mechanism of (+) supercoiled DNA binding using magnetic tweezers.Their first experiment shows that, in the presence of GapR, introduction of (+) supercoiling reduces the extension of the DNA molecule but to a lower degree than in absence of GapR. The authors conclude that GapR precludes the conversion of (+) supercoiling to (+) writhe. It is not clear though where is the excess (+) supercoiling going. Does GapR favour conversion of the added (+) supercoiling into (+) twist? Is there any clue of this in their crystallographic data? What is their model for how GapR is constraining the added (+) supercoiling? The scheme in Figure 1G is very confusing in this regard.

Our prior study (Guo et al., 2018) demonstrated that GapR specifically constrains (+) supercoiled DNA in several ways. First, we incubated purified GapR with a relaxed plasmid, treated with topo I to relax any (-) supercoils that arise if GapR introduces or stabilizes (+) supercoils, and then treated with proteinase K. The resulting plasmid was shown unambiguously by 1D and 2D chloroquine gel analysis to harbor (+) supercoils. Second, we incubated purified GapR with a nicked plasmid (an assay similar to that presented in Figure 3A of the current paper), then added ligase, treated with proteinase K, and examined the resulting plasmid by 1D and 2D gels – again, we found that GapR introduced (+) supercoils into the plasmid. Additionally, in collaboration with Maria Schumacher, we solved a crystal structure of GapR in complex with DNA, which revealed the striking tetramerization of GapR around DNA. The DNA in the structure appeared to have a narrowed minor groove and widened major groove, with helical twist estimated at 44.5º compared to 36º for B-DNA. A similar value of twist per GapR was calculated from our in vitro nicked plasmid assay. Notably, the cavity within the GapR tetramer was relatively narrow and would better accommodate overtwisted DNA than relaxed, B-form DNA.

The prior work indicated that GapR binds DNA and overtwists it in the crystal structure. The new single-molecule experiments presented in this manuscript now directly show this in solution, by showing that the main effect on the "hat curves" (extension with linking number) is a shift of their peaks to positive linking number density (Figures 1C and S1C), saturating at about σ ~ +0.015 for saturating (large) GapR concentration. Under the assumption that the DNA twisting in the crystal structure is similar to that in solution, this indicates that enough GapR binds to add linking number (in the form of twist) corresponding to σ ~ +0.015, or about 15 turns, corresponding to ~600 bp covered by GapR tetramers.

There is no appreciable compaction or shortening of the DNA upon GapR binding, indicating that at least in our single-DNA experiments there is no appreciable bending, chiral or otherwise, or stabilization of plectonemic regions (e.g., by binding crossovers, thus cross-bridging DNA and likely compacting it at low forces < 0.3 pN). Therefore, the single molecule experiments show unambiguously that GapR constrains (+) supercoiling by binding (+) twist in solution. These results were not clear from prior studies, thus justifying the presentation of the single-DNA studies. We have added mention of this to the revised manuscript.

We agree that Figure 1G (now Figure 1H) and its description was not as clear as it could have been and it has now been replaced with a clearer and revised figure and description of GapR interaction with DNA (see below).

The authors then monitor changes in DNA extension over time in absence and presence of GapR for (+) and (-) supercoiled DNA substrates. They conclude that GapR dynamically binds and diffuses along (-) supercoiled DNA but stably binds to (+) supercoiled DNA. I do not see the evidence for the former claim (dynamic binding and diffusion/sliding). I do not fully agree with the evidence shown for the second conclusion: 1 μm is a pretty high concentration for magnetic tweezers experiments. What is the binding affinity of GapR? Their moderate effect of GapR at lower concentrations (100nM) suggests that the binding affinity is likely in the hundreds of micromolars. In this range, the residence time of a protein on its substrate is short (sec) thus I am confused as to why the authors say GapR binds stably to (+) supercoiled DNA.

We are grateful for this comment which led us to clarify discussion of our model for the very clear and dramatic difference between the length dynamics for (+) and (-) supercoiled DNA following GapR binding. The dynamics are not entirely a result of GapR on/off binding dynamics. We know this because GapR is rather stably bound in the manner of other DNA-architectural proteins we have studied (e.g., HU or Fis) as shown by experiments where we wash solution phase protein away. We see the effects of GapR binding persisting in such experiments at zero protein concentration in solution (Figure 1-Figure supp. 1H-I).

Regarding the 1000 nM concentration, it is quite reasonable to see binding coming up between 100 and 1000 nM since the latter is roughly the in vivo GapR concentration. We estimate roughly 2000 to 3000 GapR copies per *BSu* cell which given the cell internal volume of roughly 2 cubic microns corresponds to just about 1 µM concentration. Other “DNA architectural” proteins found in bacteria (HU, H-NS, IHF, Fis, etc.) are found at comparable or even higher concentrations. We have added a comment to this effect to the manuscript.

Figure 1G shows the behavior of GapR-bound DNA as the DNA is wound and unwound, first for naked DNA and then for 1000 nM GapR. The "hat curve" becomes asymmetric with larger fluctuations on the negative side after the solution concentration of GapR is raised. Then, when the protein solution is replaced with protein-free buffer ("wash"), we find that the hat curve remains similar to that for the 1000 nM GapR curves (new panel Figure 1-Figure supp. 1H), with similar fluctuations (new panel Figure 1-Figure supp. 1I). This result indicates that most of the GapR that was previously bound is staying on the DNA even after removal of protein from the surrounding solution, an effect similar to what we have observed for other DNA architectural proteins with binding thresholds at similar (100-1000 nM) levels

(e.g., HU, https://pubmed.ncbi.nlm.nih.gov/15504049/ and Fis, https://pubmed.ncbi.nlm.nih.gov/21097894/). The new supplementary figure panel S1I indicates that this bound GapR is able to continue to induce large fluctuations in extension even in the absence of protein in solution.

These results indicate that the dynamical difference between the (+) and (-) supercoiled DNA with GapR present is not completely due to binding/unbinding dynamics of the GapR but is also contributed to by the dynamics of the DNA itself. This is most likely dynamics of exchange between melting and plectonemic supercoiling of DNA. Coexistence of B-DNA, melted DNA, and supercoiled DNA are well documented for negative supercoiling at slightly higher forces (typically above 0.7 pN), where the "flattening" of the hat curve for negative supercoiling is a signature of the formation of a mixture of supercoiled and melted DNA (https://pubmed.ncbi.nlm.nih.gov/9724746/, https://pubmed.ncbi.nlm.nih.gov/24606941/). GapR may also be moving around on the DNA, further contributing to the large-amplitude extension fluctuations, but evidently it is very slow to dissociate to solution (Figure 1-Figure supp. 1H-I). GapR likely is stimulating melting of DNA by staying stably bound even when the DNA is negatively twisted, since binding of GapR is driving up the negative torque (favoring melting), due to its constraint of positive twist. Therefore, we observe the "melting" behavior characteristic of negative supercoiling at lower forces (0.3 pN) than for naked DNA (> 0.5 pN). We have discussed this revision of the GapR binding model in the revised manuscript.

All in all, magnetic tweezers experiments are interesting, but do not clearly demonstrate binding of GapR to (+) supercoiling DNA, do not provide further mechanistic insight or strong support for their model of how GapR binds to (+)/(-) supercoiled DNA. Perhaps a higher resolution assay (e.g rotor bead) could shed more light into the mechanism of binding of GapR.

As indicated above we feel that the direct observation of binding and twisting of DNA in the first real-time MT experiments on GapR adds a large amount to the paper and we hope that the improvements made based on the referee suggestions will convince the reviewer of this.

Next, the authors use ChIP-seq on GapR and a fusion of GapR to 3xFLAG. They see that these proteins bind to AT-rich regions located downstream from a ribosomal operon, and their binding is reduced when transcription is inhibited. In the example shown, there are several highly transcribed ribosomal genes in the operon. I would have expected to see binding of GapR throughout the operon or at least at the 3' ends of these genes, as RNApol2 is producing (+) supercoils throughout the entire operon. However, binding of GapR appears only at the 3' end of the operon. Do these experiments really show that GapR binds to (+) supercoils in *E. coli* as claimed by the authors?

Our ChIP-seq on GapR reveals binding downstream of most highly expressed genes, not just the ribosomal operon shown in Figure 2B as one specific example. Figure 2E shows a summary of the top 300 most expressed genes in *E. coli*, revealing significant, transcription-dependent binding of GapR at the 3' ends of nearly all of these genes/operons, with Figure S2F demonstrating that GapR binding is not explained by AT-content.

We think it is actually quite difficult a priori to know or expect what to see with respect to the distribution of (+) supercoils within transcribed regions of genomes. Although the twin-domain model strongly predicts that (+) supercoils arise in front of translocating RNA polymerase (in *E. coli* there is only one RNA polymerase), whether these supercoils are retained within gene bodies or accumulate primarily at the 3' ends of genes/operons is not easily predicted nor has it been previously established. One could predict that (+) supercoils accumulate throughout an operon. However, if an operon is transcribed by multiple, closely-spaced RNA polymerases, the (+) supercoils introduced ahead of one RNA polymerase may be eliminated by the (-) supercoils that arise in the wake of the downstream polymerase. Thus, (+) supercoils may only persist beyond where the RNA polymerases terminate, which is what we generally observed in our GapR-seq studies. In short, this is where we see the value of our GapR-seq method: it now provides a powerful and rapid means of investigating and mapping the distributions of (+) supercoils in vivo! As noted earlier by the reviewer, our method, like any ChIP-seq approach, can only report on those binding events persistent enough to be captured in a population-averaged approach like ChIP-seq. So there may be accumulation of (+) supercoils transiently at other locations or at some locations in small fractions of cells. We have now amended the manuscript to clarify this issue (see p.7 of the revised text and new Figure 2A, C).

The authors then used mutants of GapR to determine if binding of GapR to DNA was due to chromosome accessibility or to DNA topology. For this they used a deletion mutant of GapR that binds DNA but does not encircle it. They show that this mutant binds (+) supercoiled DNA in a biochemical topological assay. However, it would be more consistent to use the magnetic tweezers assay of Figure 1. Does this mutant also suppress accumulation of (+) writhe? how are the dynamics of binding to (-), (+) DNA affected? These experiments could provide insight and support for their model of binding of GapR to (+) supercoiled DNA (see above).

We actually showed (see Figure 3A) that the deletion mutant of GapR (GapR^1-76^), previously found to form constitutive dimers (Lourenço et al., 2020), binds DNA but does *not* bind (+) supercoils in the in vitro topology assays, supporting the notion that tetramerization and encircling of the DNA is required for GapR to constrain (+) supercoils. And our ChIP-seq analysis of this mutant confirm that this mutant also no longer accumulates downstream of highly expressed genes in a transcription-dependent manner. Together, we think these results support the conclusion that an ability to recognize (+) supercoiled DNA enables full-length, wild-type GapR to bind, and report on the location of, (+) supercoiled DNA in vivo.

Next, they used ChIP-seq to show that the deletion mutant fails to bind to AT-rich regions at the 3' ends of highly expressed operons. From this, they conclude that binding to (+) supercoiled regions requires tetramerization. I am not convinced that the data supports this conclusion. If the tetramerization domain was required to increase the affinity of DNA binding of GapR dimers, then one would also expect a loss in specific binding in absence of the tetramerization domain. If this domain was needed for interactions with other proteins present at 3'-ends of highly transcribed genes, then one would also expect a similar result.

As shown in Figure 3-Figure supp. 1B, the GapR^1-76^ mutant leads to band shifting in EMSAs at approximately the same concentration as the wild-type GapR, though of course not with the same banding pattern given that the mutant cannot encircle DNA like the wild-type protein. This result indicates that tetramerization likely is not required for a major increase in affinity. We also find that the GapR^1-76^ mutant binds DNA in vivo, as indicated by the presence of discrete peaks in the ChIP-seq analysis (examples are shown in Figure 3B and S3D), but it no longer localizes downstream of highly expressed genes/operons, especially convergent genes/operons, as seen with the wild-type GapR (see Figure 3C and S3D-E). Finally, we think the likelihood of the deleted region of GapR promoting binding to another protein present at the 3' ends of highly transcribed genes is very low given that GapR homologs are only found in α-proteobacteria, which are not closely related to *E. coli*. Moreover, we also see GapR binding downstream of highly transcribed genes in the eukaryote, *S. cerevisiae*, which is exceedingly unlikely to have a binding partner of GapR that specifically recruits it to the regions where we see occupancy in ChIP-seq.

Reviewer #2:This work claims to provide a new tool that specifically detects positive DNA supercoiling, genome-wide. The research question asked in the manuscript is quite important. The tool is based on the ability of bacterial protein GapR to bind preferentially with over-twisted DNA. The manuscript is divided into two parts: first, a demonstration that bacterial protein GapR binds preferentially with over-twisted DNA, both in single-molecule assays and in vivo at sites of already predicted to be positively supercoiled, and then a genome-wide search for positive supercoiling at the key genomic positions. There are several promising results in the second part of the manuscript, but they are all based on the first part where the presented evidence/data are not sufficient to draw decisive conclusions. Many additional experiments, much more data and further controls are required to prove that GapR could be used as a probe for positive DNA supercoiling.1) The authors in their previous work used analysis of the DNA supercoiling induced by plasmid-GapR interaction to suggest that the protein likely binds over-twisted DNA. In the current manuscript, Guo et al., perform again this analysis (Figure 3A, S3E). The weak point in their DNA supercoiling assay is that DNA topology does not change up to 2.5 μM GapR, and then an abrupt change is observed as the concentration is increased. This is not the expected pattern if GapR binding increases incrementally as it introduces or stabilizes a small amount of positive twist in the DNA (see Clark and Leblanc, Methods Mol Biol. 2015 for a recent review of this method). The distribution of topoisomers in the assay should gradually shift from the relaxed state to the new supercoiled position until binding is saturated. One of the simplest explanations of the observed unusual pattern is a synergy threshold: for example, the GapR-mediated DNA bridging which is reported in the literature (see Lourenço et al., mBio. 2020) might give this result. To confirm their suggestion, the authors use single-molecule assays. Based on the pattern of DNA "rotation-extension curve" generated by this assay, the authors state that GapR stably interacts with positively supercoiled DNA while the interaction with negatively supercoiled DNA is unstable. However, as admitted in the manuscript this curve is highly unusual and cannot be explained solely by the constraining of positive supercoils. The high fluctuations of DNA length on the negative-supercoiling side of the curve once again suggests that some kind of cooperative binding-unbinding of GapR affects the shape of the DNA.

We have now performed the same plasmid topology assays at many more intermediate concentrations. As the reviewer anticipated, the distribution of topoisomers shifts gradually from relaxed to supercoiled, with the 2D gels again confirming again that GapR is binding and stabilizing (+) supercoils. These new data are presented in the revised manuscript (see new Figure 1A, S1B).

As discussed above, for negative supercoiling, it is likely that GapR-DNA interactions are quite stable (based on the wash experiments now included in the supplement, Figure 1-Figure supp. 1H-I) and are not by themselves the explanation of the large fluctuations. Instead, it appears more likely that the large fluctuations are exchanges between strand-separated DNA (large negative σ and long length) and plectonemic supercoils (also negative σ but short length). The combination of unwinding and GapR binding (which constrains positive twist) causes unwinding-driven melting to occur at lower forces than for naked DNA. In addition to this, GapR may be able to move around on DNA without dissociating to the bulk, providing another mechanism to cause extension fluctuations while remaining associated with the DNA.

2) The authors imply that the GapR-binding method might be superior to psoralen-crosslinking methods for the detection of positive supercoiling but there is no actual comparison. Psoralen assays have been calibrated both in vitro and in vivo (see Bermúdez et al., Nucleic Acids Res. 2010, and Kouzine et al., Nat Struct Mol Biol. 2008). Similar calibration is required for GapR study. In the current manuscript, the authors detected GapR binding at sites expected to be positive supercoiling (Figure 2, 3, and 4) which is not sufficient to support the key claims in the manuscript – GapR is binding at positively supercoiled sites. Based on the known topological plasticity of chromatin to the DNA over-twisting, one might expect that GapR is able to differentiate between difference positive torsional stress stored in twist with that in the writhe of the 3-D shape of DNA in the E. coli genome or in yeast chromatin. Although exciting, the full characterization of DNA-GapR interaction is required.

Again, as detailed in responses above, we think the results presented previously in Guo et al. 2018 and in this paper, including the new results in Figure 1 and S1, strongly support the notion that GapR specifically recognizes overtwisted DNA. This work supports the extensive ChIP-seq studies previously done for *C. crescentus* and now for *E. coli* and *S. cerevisiae*, which indicate that GapR localizes to regions of (+) supercoiled DNA in vivo. With regard to calibrating GapR-seq like what has been done for psoralen-based studies, we would point out that the latter studies are based on preferential intercalation of psoralen and its derivatives with (-) supercoiled DNA, so the more supercoiling, the more psoralen crosslinking can be detected. The situation is different with GapR in the sense that it recognizes overtwisted DNA, which, for a given stretch of DNA, isn't something that can quantitatively vary. In other words, DNA can either be overtwisted such that GapR can bind it, or not. As a result, we do not think it is appropriate to relate the extent of GapR binding to a quantitative value of (+) superhelicity.

3) The strength of the manuscript is the technically impressive analysis of the GapR localization in the genome reported in the second part of the manuscript. The authors find that this protein does recognize the strategic regions of the genome (Figure 6 and 7). With proper analysis of DNA-GapR interaction in the first part of the manuscript, these data will indicate that GapR is an important probe for DNA conformation in the context of key genomic processes.

We feel that the responses above (see, in particular, the response to point #1 from Reviewer 1) summarizing the prior work on GapR combined with the additional MT data presented here provide convincing evidence that GapR does indeed bind (+) supercoils by recognizing overtwisted DNA and, consequently, that it can in fact be used as a useful and powerful probe for DNA conformation. We think the results in Figures 6 and 7, as noted by the reviewers, suggest that GapR-seq can be used to provide important new insights into a variety of genomic processes.

Comments for the authors:1) The quality of the 2D gels should be improved and accurate titration with broader range of protein concentrations should be performed. The explanation of the topoisomers' distribution should minimize pre-assumptions. The ability of the protein (protein preparation) to induce DNA double-stranded breaks and nicks should be explained.

As noted above, we have now performed additional 1D and 2D gels for the analysis of plasmid topology at many more concentrations of GapR. This preparation of GapR does contain a trace nuclease activity, which, as the reviewer noted, can lead to formation of double-strand breaks. This nuclease activity is likely the benzonase that was added at the cell lysis step to degrade DNA and increase recovery of GapR protein. We have now performed a no ligase control experiment (new Figure 1-Figure supp. 1A) to show the behavior of the nuclease. As noted in our previous work (Guo et al., 2018), this nuclease activity cannot explain the topological changes to plasmids mediated by GapR in vitro or its effects on DNA topology in the single-molecule assay for multiple reasons. First, ligation of degraded DNA (the DNA smear) will lead to small circles of varying size, rather than the larger, specific topoisomers observed in the topology assays. Second, the nuclease effects are only observed at 0.5-2.0 μm GapR, which does not explain the changes in topology that can be observed at > 2.0 μm GapR (Figure 1A, 3A). Lastly, benzonase is dependent on divalent cations, and its activity is stimulated in the ligation reaction due to the high concentration of Mg^2+^. In the absence of Mg^2+^, which is not present in the binding buffer for MT, there is no nicking activity. This can be observed in Author response image 1. We now discuss this activity in the Methods.

**Author response image 1. sa2fig1:** Comparison of contaminating nuclease in GapR prep in the presence or absence of magnesium. At high concentrations of GapR, GapR binding to plasmid DNA protects DNA from nuclease degradation.

2) The DNA topology electrophoresis and single-molecular assays were performed at different protein concentrations. What is the reason for choosing different concentrations? What would we see if single-molecular assay would be performed at higher protein concentration? All anomalies on the extension curve should be explained (might be added to the Supplementary section). What is the reason for the high fluctuation on the negative side of the curve? I do not think it can be explained by single binding-unbinding-diffusing event. Why is naked, relaxed DNA is shorter than the same DNA in the presence of the protein? One might expect the opposite if GapR constrains ower-twisted DNA.

We carried out the MT experiments over a concentration range where effects were observed on the rotation-extension and force-extension experiments, which happens to be not far from the in vivo concentration (see above). At higher protein concentration (more than 1 μM) we may well start to run into issues regarding the GapR affecting adhesion of DNA to the tethering surfaces. The lengthening of the DNA at 1 μM GapR concentration is quite interesting, and suggests that the GapR-DNA complex overtwists and lengthens dsDNA. While this may seem counterintuitive, in fact the "natural" initial response of DNA to overtwisting is in fact lengthening (see https://pubmed.ncbi.nlm.nih.gov/16712339/ and https://pubmed.ncbi.nlm.nih.gov/16862122/). GapR may take advantage of this tendency to introduce simultaneous DNA lengthening and overtwisting – also see our response above to related questions in point #1 from Reviewer 1.

Furthermore, in response to the concentration-dependent topology issues raised above by this reviewer, we now show with additional 1D and 2D assays that the GapR-dependent effects on topology can be observed at the same concentrations as is used for the MT experiment (1 μM GapR, Figure 1A, S1B).

3) Preferential binding of GapR to positively supercoiling DNA over negatively supercoiled DNA was not proven as the single-molecular assay did not give a definitive answer. The study should be supplemented by competition assay between different DNA conformations. Could you efficiently fish out positively supercoiled plasmid from the mixture of genomic DNA circles/plasmids wound to different degrees? What happens if you compare normal plasmids (able to form supercoils) with DNA minicircles (unable to form supercoils)?

As noted in several of the responses above to both reviewers, we think the magnetic tweezer data, the first to be reported for GapR, together with the prior topology studies and GapR-DNA crystal structure, do yield a clear picture of the preferential binding of GapR to overtwisted DNA.

4) In the omics study, GapR binding should be compared with psoralen-based maps. The assumption that psoralen-based studies infer the presence of positive supercoils by the absence of crosslinking is wrong. In the classical approach, the presence of supercoiling is inferred from the changes of psoralen intercalation after fast nicking and relaxation of the DNA inside the cells (Sinden's studies). In addition, it is incorrect to say that "psoralen-based studies are still limited in resolution". With developing of high throughput sequencing methods, the resolution of supercoiling is close to the DNA persistence length (Henikoff's studies).

We thank the reviewer for raising this important point about comparing GapR-seq and psoralen methods. We now provide a direct comparison of GapR-seq and psoralen microarray data for *S. cerevisiae* (from Achar et al., 2020) in a series of new figures (Figure 5-Figure supp 1H-I, Figure 6-Figure supp. 1A). We find that there is no correlation between GapR enrichment and psoralen de-enrichment (new Figure 5-Figure supp 1I). Most notably, the psoralen-based work (Achar et al., 2020) suggested that (+) supercoils accumulate within gene bodes while (-) supercoils accumulate downstream, and there is no correlation between supercoiling intensity and transcriptional strength (see Author response image 2). These findings are, in our view, in conflict with the twin-domain model. In contrast, our GapR-seq indicates that (+) supercoils arise most clearly at the 3' ends of highly expressed genes in a transcription-dependent manner, which we believe is fully consistent with the twin-domain model of supercoiling. Further, we find that regions suggested previously and in this work to harbor (+) supercoils, such as centromeres, cohesin-binding sites, the rDNA loci, ARS sequences, and R-loop adjacent regions, were not found to have (+) supercoils using psoralen immunoprecipitation. We now show examples of psoralen-based maps at centromeres (Figure 6-Figure supp 1A), as there is significant previous literature that suggests (+) supercoiling is associated with centromeric sequences and our GapR-seq results support this idea.

We have removed the statement about (+) supercoils only being inferred by the absence of crosslinking – we agree that the statement was incorrect as written. With regard to resolution, the original statement was based on our understanding that psoralen methods utilize very sparse crosslinking (1-2 per 10 kb), which seems likely to limit the resolution, and some of the prior studies relied on oligo tiling arrays with potentially more limited resolution than sequencing assays. Nevertheless, we have removed this statement. There are strengths and limitations of both GapR-seq and psoralen-based methods. We think the field will ultimately benefit from the development, characterization, and comparison of multiple, independent methods, a point we now make more clearly in the revised manuscript.

**Author response image 2. sa2fig2:** Correlation between mean psoralen (mean(bTMP IP/input)) and transcriptional strength (log_10_(rpkm)) of all expressed *S. cerevisiae* genes.

5) All discussion of the positive supercoiling in yeast should be supplemented with the introduction of known torsional plasticity of chromatin to DNA over-twisting. The current consensus in the field is that chromatin fiber is torsionally soft with respect to positive supercoiling – twisting of chromatin results in the reorganization of nucleosomal template without introducing DNA over-twisting. How do you align this expected topological plasticity with the proposed ability of GapR to sense twist rather than writhe?

GapR binds to positively twisted DNA and can be a potent source of constraint of linking number. In a topologically intact chromatinized plasmid or a chromatin loop, GapR binding will occur, generate positive twist where it binds DNA, and generate negative compensating linking number change in the remainder of the DNA. The extent to which GapR binds will depend on how positive the DNA torque was before binding, which in turn probes the degree of positive supercoiling. We agree that whether the DNA is chromatinized or otherwise protein-bound will affect its linking number modulus and whether linking number change will tend to end up in twist or in writhe. The referee’s comment that chromatin is soft to positive twist and that positive linking number change will lead to nucleosome reorientation (most likely via relatively low-free-energy changes in DNA writhe through DNA unpeeling from nucleosomes combined with chromatin fiber bending) is highly relevant to understanding the conformation of positively supercoiled chromatin. However, regardless of initial conformation, if there is initially positive torque in the DNA, it will reduce the free energy of binding of GapR to DNA, independent of whether the chromatin has responded to that positive torque via twist or by writhe. In either case the initial positive linking number constraint represents stored free energy that will reduce the free energy of GapR binding. We now discuss this in the manuscript (see p. 19).

6) There is very little discussion on consequences of the prolonged expression of GapR protein. So, one caveat is that the expression of GapR over time does not perturb the DNA topology or chromatin conformation in cells. There may be quick and more general approaches for this method than building strains and cell-lines to express GapR. For example, the authors could make yeast spheroplasts, treat with saponin/digitonin, in the presence of GapR, fix with formalin and then perform ChIP-seq, (this is the sort manipulation that is used in native ChIP) and then compare these results with in vivo expression of GapR. This would potentially eliminate artefacts of prolonged in vivo expression and greatly expand the general utility of the method as it could be than used on any cell line without transgenic or knock-in expression.

We thank the reviewer for their suggestion of ways to potentially improve or modify the use of GapR to probe positive supercoiling in yeast. Implementing these ideas and validating them will, however, require substantial amounts of work that are beyond the scope of this paper at hand. As with any method, we hope our initial report spurs subsequent modifications and improvements. However, to directly address the reviewer’s concern that expression of GapR perturbs DNA topology, we have now examined transcriptional changes upon GapR expression in yeast to determine if cellular stress responses including DNA damage and replication stress responses are activated or if supercoiling is perturbed. We do not observe significant change in supercoiling-sensitive transcripts (< ~2-fold change in *PHO5*, *GAL7*, *GAL10*, *ADH2* (Pedersen et al., 2012)). Furthermore, we do not observe significant induction of major stress responses (< ~2-fold change in Pka1-, Hog1-, Hsf1-, and Yap1-dependent gene expression), the unfolded protein response (< ~2-fold change in *HAC1*/*BIP* levels) and DNA damage responses (< ~2-fold change in genes regulated by checkpoint kinases Mec1, Tel1, Chk1, Rad53, and Dun1 (Jaehnig et al., 2013)). We now present this analysis in the manuscript and in new Figure 4-Figure supp. 1B.

7) The sentence "Genomic DNA can become supercoiled when the DNA duplex winds about its own axis forming a right-handed superhelix (positive supercoiling) or a left-handed superhelix (negative supercoiling)" is misleading. In the plectonemic form of unconstrained supercoiling, a right-handed superhelix is assigned a negative number (negative supercoiling) and a left-handed superhelix is assigned a positive number (positive supercoiling). Opposite for solenoid/toriodal form of constrained supercoiling.

This is absolutely correct and we have corrected this mistake. No major results of the paper were affected by this admittedly dumb error.

8) Moderate binding is detected to 5' ends of weakly expressed genes but not to highly expressed gene. It would be good to check if this is due to an upstream co-directional transcript that pumps positive supercoils into the promoter of the weakly expressed genes making the promoter harder to melt and transcribe.

This is a great point that the reviewer has raised. We have added a new analysis in *S. cerevisiae* correlating intergenic GapR enrichment with the expression level of the upstream (5') gene or the downstream (3') gene for co-oriented genes. We find that GapR enrichment is correlated with the expression of the upstream gene and uncorrelated with the expression of the downstream gene (Figure 4-Figure supp. 1K). A correlation between GapR enrichment and upstream gene expression levels were also found when the downstream gene was poorly expressed (Figure 4-Figure supp. 1K). We conclude, as the reviewer hypothesized, that positive supercoiling at the 5' end of poorly expressed genes is generated by the upstream co-directional transcript. This idea has been integrated into the revised text on p. 10.

9) DNase likes G-C rich DNA better than A-T-rich, GapR likes the opposite. Does the base composition partly explain separation of GapR and DNaseI sites?

Although both GapR and DNase have modest nucleotide preference, these biases cannot account for the separation of GapR and DNase sites that we observe. When we examined the DNase-seq dataset, we observed many single positions that had extremely high representation in DNase-seq, likely due to DNase-sequence specificity or from other sources of library bias. To decrease sequence biases and because we were interested in GapR-accessible chromatin, we restricted our comparison between GapR enrichment and DNase enrichment (Figure 5-Figure supp. 1G) to DNase accessible regions significantly longer than a GapR binding site (DNase sites > 50 bp). These long, DNase-accessible regions have very modest GapR enrichment. Similar results were obtained for shorter DNase-accessible regions (i.e. DNase sites >10 bp, >2 bp and all regions). We have now directly examined the GC content of the DNase-accessible regions from Figure 5-Figure supp. 1G and GapR-enriched regions from Figure 5E, finding that these regions are both modestly AT-rich. We now include this analysis in Figure 5-Figure supp. 1H and have amended the manuscript and methods to clarify this issue.

10) The idea of involvement of positive supercoiling in R-loop genesis should be discussed together with recent work from Chedin' lab (Stolz et al., Proc Natl Acad Sci U S A. 2019).

We now cite and discuss the Stolz et al., 2019 paper in the section related to R-loops.